# CAVINR: Coordinate-Aware Attention for Video Implicit Neural Representations

## Abstract

Implicit Neural Representations (INRs) have emerged as a compelling paradigm, with Neural Representations for Videos (NeRV) achieving remarkable compression ratios by encoding videos as neural network parameters. However, existing NeRV-based approaches face fundamental scalability limitations: computationally expensive per-video optimization through iterative gradient descent and convolutional architectures with shared kernel parameters that provide weak pixel-level control and limit global dependency modeling essential for high-fidelity reconstruction. We introduce CAVINR, a pure transformer framework that fundamentally departs from convolutional approaches by leveraging persistent cross-attention mechanisms. CAVINR introduces three contributions: a transformer encoder that compresses videos into compact video tokens encoding spatial textures and temporal dynamics; a coordinate-attentive decoder utilizing persistent weights and cross-attention between coordinate queries and video tokens; and temperature-modulated attention with block query processing that enhances reconstruction fidelity while reducing memory complexity. Comprehensive experiments demonstrate CAVINR's superior performance: 6-9 dB PSNR improvements over state-of-the-art methods, $10^5\times$ encoding acceleration compared to gradient-based optimization, $85-95\%$ memory reduction, and $7.5\times$ faster convergence with robust generalization across diverse video content, enabling practical deployment for large-scale video processing applications.

## 1 Introduction

Video representation poses a significant challenge in computer vision due to the substantial computational and storage requirements of high-dimensional video data. The growth of video content across streaming platforms, autonomous systems, and multimedia applications has created demanding requirements for efficient video representation and processing that are far more complex than static image processing. Traditional video compression standards like H.264/AVC Wiegand et al. (2003) and HEVC Sullivan et al. (2012) use handcrafted codecs that struggle to balance compression ratios with reconstruction quality, while requiring substantial computational resources for high-resolution video processing. Recent research has investigated representing videos as Implicit Neural Representations (INRs) Chen et al. (2021a); Li et al. (2022b); Kim et al. (2022), where videos are encoded as neural network parameters, enabling compact storage while supporting downstream tasks such as super-resolution and denoising. The NeRV series Chen et al. (2021a; 2022a; 2023) Introduced this approach by using frame index as input to convolutional networks to generate frames, achieving significant speed improvements over coordinate-based methods while maintaining competitive compression ratios and visual quality. However, NeRV-based methods face important scalability challenges that limit practical deployment. The primary limitation comes from computationally expensive per-video optimization through iterative gradient descent, making encoding costly for large-scale applications. Since videos are encoded once but reconstructed repeatedly during playback and processing, both reconstruction quality and inference speed are important performance factors. Recent acceleration efforts through MetaNeRV Guo et al. (2025) and FastNeRV Chen et al. (2024) use meta-learning and transformer-based hypernetworks, but remain limited by the inherent constraints of convolutional operations. Shared kernel parameters provide limited pixel-level control, while local connectivity prevents effective global dependency modeling, resulting in suboptimal video quality, slow convergence, and insufficient reconstruction fidelity for demanding applications.

We present **CAVINR** (Coordinate-Aware Attention for Video Implicit Neural Representation), a transformer-based framework that addresses the limitations of convolutional approaches by using cross-attention mechanisms with persistent parameters. The method creates direct correspondences between compressed video representations and spatial coordinate queries, improving computational

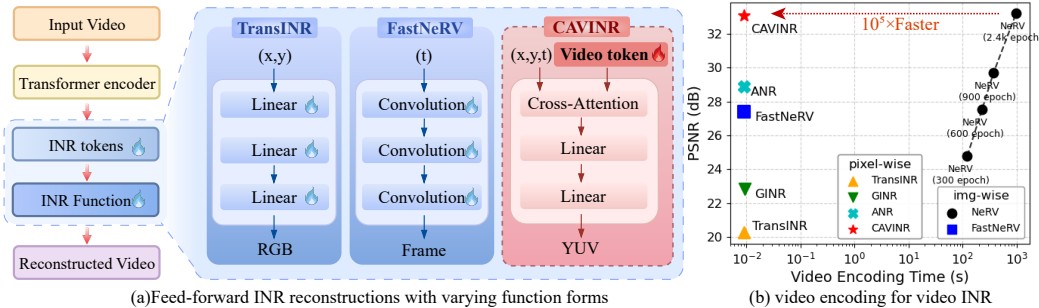

(a)Feed-forward INR reconstructions with varying function forms   (b) video encoding for video INR

Figure 1: **Framework Comparison and Encoding Speed.** (a) Feed-forward INR approaches: TransINR and FastNeRV generate video-specific parameters requiring independent weight replacement, while CAVINR employs shared transformer weights with video-specific tokens for efficient cross-attention reconstruction. (b) Encoding efficiency: CAVINR achieves $10^5 \times$ faster encoding than gradient-based NeRV Chen et al. (2021a) baseline through feed-forward processing.

efficiency and enabling precise pixel-level control through the global modeling capabilities of transformer architectures. The CAVINR framework consists of three main components: a **transformer encoder** that generates compact *video tokens* encoding spatial texture information and temporal dynamics, a **coordinate decoder** with fixed weights that applies cross-attention between coordinate queries and the video token representations, and an **adaptive attention module** that uses temperature scaling and block-based query processing to improve reconstruction performance while maintaining memory efficiency. Figure 1 compares our approach with existing methods. While conventional techniques require video-specific weight generation, CAVINR uses shared transformer weights with video-specific tokens through cross-attention, achieving $10^5 \times$ faster encoding than gradient-based NeRV with better reconstruction quality. The contributions of this work are summarized as follows:

- We propose the CAVINR architecture for learning video implicit neural representations, achieving a $10^5 \times$ speedup in encoding compared to conventional gradient-based optimization methods while delivering superior reconstruction quality.

- We introduce a coordinate-attentive decoder with persistent weights and temperature-modulated attention, establishing direct correspondences between video tokens and spatial coordinates for both computational efficiency and reconstruction fidelity.

- We design comprehensive architectural innovations including a convolution-based tokenizer, axis-adaptive position encoding, and temperature-modulated cross-attention that collectively enhance spatial-temporal modeling capabilities and representation accuracy.

- Comprehensive experiments demonstrate $6 - 9$ dB PSNR improvements over existing methods, $85 - 95\%$ memory reduction, and $7.5 \times$ faster convergence with consistent performance across diverse video content.

## 2 RELATED WORK

**Implicit Neural Representations.** Implicit neural representations offer a compact approach to signal encoding, storing images, and videos directly within neural network parameters Dupont et al. (2021); Chen et al. (2022b). The foundational coordinate-based methods Tancik et al. (2020); Sitzmann et al. (2020) pioneered this field by using multilayer perceptrons to map spatial-temporal coordinates to signal values, demonstrating remarkable performance in applications such as novel view synthesis Mildenhall et al. (2020) and image super-resolution Chen et al. (2021b). Building on this foundation, NeRV Chen et al. (2021a) proposed frame-wise implicit representations that generate entire frames directly from temporal indices through convolutional architectures. Several follow-up works have improved reconstruction quality through various strategies: E-NeRV Li et al. (2022c) applies spatial-temporal decomposition, HNeRV Chen et al. (2023) combines hybrid variational autoencoders. Despite these advances, coordinate-based methods still achieve superior representation accuracy in many scenarios Chen et al. (2022c); Kim et al. (2022); Aiyetigbo et al. (2025).

**Hypernetwork-Based Video INR Representations.** Hypernetworks Ha et al. (2017) offer a flexible framework for generating adaptive model parameters based on input data. Early work in neural representations explored weight modulation using latent vectors Park et al. (2019); Mescheder et al.

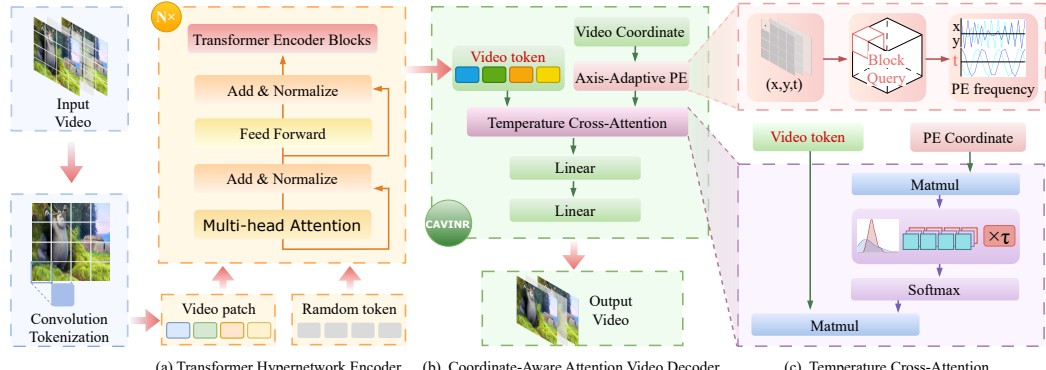

Figure 2: CAVINR Framework. (a) The Transformer Hypernetwork Encoder extracts compact video tokens from input video sequences. (b) The Coordinate-Aware Attention Video Decoder maps spatiotemporal coordinates to pixel values by first applying Axis-Adaptive Position Embedding to coordinate queries, then using (c) Temperature-Modulated Cross-Attention between position-embedded coordinates and video tokens to reconstruct the final video.

(2019), while recent approaches employ transformer architectures to directly produce parameters for implicit neural representations. Several methods follow this paradigm, including TransINR Chen & Wang (2022), GINR Kim et al. (2023a), FastNeRV Chen et al. (2024), and ANR Zhang et al. (2024), which utilize transformer-based hypernetworks for content-adaptive weight generation. These techniques build upon convolution concepts Chen et al. (2020); Yang et al. (2019) to enable instance-specific parameter synthesis. However, existing hypernetwork methods rely on convolutional priors that constrain their capacity to model global dependencies. Our approach addresses this limitation by integrating persistent decoder weights with transformer-based cross-attention mechanisms.

**Neural Video Compression.** Traditional video codecs such as H.265/HM HM, H.266/VTM VTM, and ECM ECM remain widely used in current applications, though they suffer from high computational complexity and limited compression efficiency. Learning-based compression methods Rippel et al. (2019); Agustsson et al. (2020); Maiya et al. (2023) achieve better rate-distortion performance but introduce significant decoding latency that limits their practical deployment. Neural video coding has evolved along two distinct paradigms. The DCVC family Li et al. (2021); Sheng et al. (2022); Li et al. (2022a; 2023; 2024); Jia et al. (2025) represents compression-oriented methods that prioritize bitrate efficiency and real-time coding speed for transmission applications, achieving competitive performance with traditional codecs through conditional coding and entropy modeling. Implicit neural representations offer an alternative by encoding videos as neural network parameters. NeRV Chen et al. (2021a) introduced this concept, demonstrating competitive performance through model compression while supporting GPU-accelerated decoding. DNeRV He et al. (2023) incorporates entropy coding while preserving speed benefits. These representation-oriented methods prioritize reconstruction quality and downstream task support over bitrate optimization. Building on the representation paradigm, our approach targets quality-first applications by introducing coordinate-attentive decoding and memory-efficient attention mechanisms, achieving superior reconstruction fidelity within practical computational constraints.

**Temperature-Modulated Attention Mechanisms.** Temperature scaling provides an effective approach for controlling the sharpness of attention distributions in various machine learning tasks. In natural language processing, Zhang et al. Zhang et al. (2021) showed that attention smoothing improves abstractive summarization, while SACT Lin et al. (2018) enhanced machine translation quality using self-adaptive temperature scaling. Computer vision tasks have also benefited from temperature modulation. Zhou et al. Zhou et al. (2023) improved image inpainting by using temperature-scaled attention to better leverage contextual information. We apply *temperature-modulated cross-attention* to coordinate-based video reconstruction. While previous work has focused on self-attention mechanisms, our method adjusts temperature parameters in cross-attention to better optimize interactions between coordinate queries and video tokens. This design enhances reconstruction quality while preserving computational efficiency, addressing the core challenge in neural video representation: precise coordinate-content alignment for high-quality decoding.

## 3 METHODS

### 3.1 PROBLEM STATEMENT

Let $V \in \mathbb{R}^{T \times C \times H \times W}$ denote a video sequence of length $T$, where each frame contains $C$ channels with spatial resolution $H \times W$. The objective of video implicit neural representation (INR) is to construct a parametric function $f_\theta$ that encodes the entire video $V$ within its learned parameters $\theta$.

NeRV-based networks learn a direct mapping from frame index to corresponding RGB images, enabling video reconstruction through:

$$\hat{V}_t = f_\theta(t) \quad \forall t \in [1, T] \tag{1}$$

where $\hat{V}_t$ denotes the reconstructed frame at time $t$, and the optimized weights $\theta$ constitute the implicit video representation that parametrically encodes the complete visual content.

However, existing implicit neural representations can only encode a single video per model, which significantly limits their practical use. To improve encoding efficiency in video INR, Fast-NeRV Chen et al. (2024) proposes a hypernetwork $g_\phi$ that generates parametric weights $\theta' = g_\phi(V)$ directly from input video data. These generated weights are then loaded into the NeRV decoder $f_\theta$ for video reconstruction. Although FastNeRV substantially reduces video encoding time, it has important limitations that lead to reduced reconstruction quality in the decoded outputs.

### 3.2 OVERALL WORKFLOW

We propose a pure transformer architecture that addresses both efficiency and accuracy bottlenecks in neural video representations through synergistic encoding-decoding co-design. Our framework improve conventional hypernetworks through two key components: (1) a transformer encoder $g_\phi$ that compresses input video $V$ into compact latent tokens $\mathbf{T}_v = g_\phi(V)$ via spatiotemporal patch aggregation, and (2) a weight-static decoder $f_\psi$ that reconstructs frames $\hat{V} = f_\psi(\mathbf{T}_v, \Omega)$ through cross-attention mechanisms conditioned on coordinate queries. Reconstruction operates within the continuous spatiotemporal coordinate space:

$$\Omega = \{(x, y, t) \mid 0 \le x \le W, 0 \le y \le H, 0 \le t \le T\}. \tag{2}$$

Position embeddings for each coordinate $(x, y, t) \in \Omega$ serve as queries in the cross-attention mechanism. The cross-attention layer retrieves relevant information from video tokens $\mathbf{T}_v$, then processes these signals through instance-agnostic MLP layers to generate pixel values. Both the cross-attention layer and MLP use persistent weights shared across all videos, enabling efficient decoding without per-video optimization.

As shown in Figure 2, our framework enables three fundamental advances: (1) end-to-end optimization through reconstruction loss minimization $\mathcal{L}(V, \hat{V})$ with generalization across video instances, (2) decoding across all spatiotemporal coordinates via persistent network weights, and (3) enhanced reconstruction fidelity through pixel-level control. The training objective minimizes the mean squared error between the original and reconstructed videos:

$$\mathcal{L} = \frac{1}{|\Omega|} \sum_{(x,y,t) \in \Omega} \|V_{x,y,t} - \hat{V}_{x,y,t}\|_2^2 \tag{3}$$

### 3.3 TRANSFORMER HYPERNETWORK ENCODER

Inspired by generalizable INR methods for images Kim et al. (2023b); Zhang et al. (2024) and videos Chen et al. (2024), we employ a transformer network with $L$ encoder layers as a hypernetwork $g_\phi$ to generate compact video representations. Our approach introduces a learnable convolutional video tokenizer that replaces explicit patch extraction with learned spatial abstractions.

**Convolutional Tokenization** Unlike conventional methods employing unfold operations with linear projections, our convolutional tokenizer directly incorporates spatial abstraction during tokenization. The input video $V$ generates initial patch tokens $\mathbf{P} \in \mathbb{R}^{N_p \times d}$, where $N_p$ indicates spatiotemporal patch count and $d$ the token dimension.

**Token Compression** The transformer processes a set of randomly initialized learnable tokens $\mathbf{R} \in \mathbb{R}^{N \times d}$ through successive attention layers to produce compressed video tokens $\mathbf{T}_v \in \mathbb{R}^{N \times d}$:

$$\mathbf{T}_v = \text{Softmax}\left(\frac{\mathbf{Q}\mathbf{K}^\top}{\sqrt{d}}\right)\mathbf{V}, \quad \text{where} \quad \begin{cases} \mathbf{Q} = \psi(\mathbf{R}) \\ \mathbf{K}, \mathbf{V} = \psi(\mathbf{P}) \end{cases} \tag{4}$$

where $\psi$ denotes learned linear projections, and $N \ll N_p$.

This design offers several advantages over previous methods: it provides pixel-level control and local connectivity for better reconstruction quality, enables dynamic token allocation based on attention mechanisms, and creates a unified representation of spatial and temporal features through progressive compression. The resulting position-aware tokens $\mathbf{T}_v$ serve dual roles as content descriptors and reconstruction operators, supporting stable cross-attention operations with fixed transformer weights during decoding.

### 3.4 Coordinate-Aware Attention Video Decoder

Unlike conventional hypernetworks Chen & Wang (2022); Chen et al. (2024) that suffer from computational redundancy through layer-specific weight generation, our architecture establishes direct video correspondences via cross-attention mechanisms. Our persistent decoder reconstructs video frames through coordinate-to-token attention, transforming latent tokens $\mathbf{T}_v \in \mathbb{R}^{N \times d}$ into pixel values via unified spatiotemporal mapping.

**Axis-Adaptive Positional Encoding** The decoding process begins with multi-frequency positional encoding of normalized coordinates. Given input coordinates $\mathbf{x} = (h, w, t) \in [0, 1]^3$, we implement stratified frequency projection $\gamma : \mathbb{R}^3 \to \mathbb{R}^{6k}$:

$$\gamma(\mathbf{x}) = [\cos(\pi \mathbf{v}) \,\|\, \sin(\pi \mathbf{v})] \tag{5}$$

where the intermediate vector $\mathbf{v} \in \mathbb{R}^{3k}$ is computed through coordinate-wise frequency modulation:

$$\mathbf{v} = \mathbf{x} \otimes \mathbf{w} \in \mathbb{R}^{3k} \quad \text{and} \quad \mathbf{w} = \left[\sigma^{\frac{i}{k-1}}\right]_{i=0}^{k-1} \in \mathbb{R}^k \tag{6}$$

Here, $k$ specifies the number of frequency components per coordinate dimension, $\sigma$ is the frequency scaling factor controlling the wavelength range, and $\otimes$ denotes element-wise multiplication broadcasted across coordinate dimensions.

Our decoder employs axis-adaptive spectral encoding to address the intrinsic disparity between spatial and temporal video dimensions. Unlike conventional coordinate encoding methods that apply uniform frequency distributions, we implement differentiated frequency allocation:

$$\mathbf{v} = [h \otimes \mathbf{w}_s, w \otimes \mathbf{w}_s, t \otimes \mathbf{w}_t] \tag{7}$$

where the spatial-temporal frequency vectors are defined as:

$$\mathbf{w}_s = [\sigma_s^{i/(k_s-1)}]_{i=0}^{k_s-1} \text{ with } k_s = \lfloor 4k/3 \rfloor, \mathbf{w}_t = [\sigma_t^{i/(k_t-1)}]_{i=0}^{k_t-1} \text{ with } k_t = \lfloor k/3 \rfloor \tag{8}$$

This spectral stratification principle allocates higher frequency components to spatial dimensions for edge preservation while using lower frequency components for temporal encoding, addressing the spatial-temporal frequency disparity in video data.

**Cross-Attention Reconstruction** The encoded coordinates interact with video tokens through cross-attention:

$$\mathbf{F} = \text{Softmax}\left(\frac{\mathbf{Q}\mathbf{K}^\top}{\sqrt{d}}\right)\mathbf{V}, \quad \begin{cases} \mathbf{Q} = \phi(\gamma(\mathbf{x})) \\ \mathbf{K}, \mathbf{V} = \phi(\mathbf{T}_v) \end{cases} \tag{9}$$

where $\phi$ represents learnable projections establishing dynamic content-coordinate correlations. The resultant feature vector $\mathbf{F}$ undergoes nonlinear refinement through a shallow MLP:

$$\hat{v} = \text{MLP}(\mathbf{F}) \tag{10}$$

Crucially, all decoder parameters remain static across video instances, enabling: (1) parallel processing of coordinate grids for arbitrary resolutions, (2) joint modeling of local textures and global motion, and (3) hardware-friendly memory access patterns through weight persistence.

**Block Query Processing** To address the quadratic memory bottleneck in attention mechanisms, we implement chunked query processing that restricts each coordinate query's attention to local $M \times M$ windows, reducing memory complexity from $\mathcal{O}(H * W * T)$ to $\mathcal{O}(M^2)$ where $M \ll H$. This spatial-temporal constraint leverages video coherence through non-overlapping sliding windows, maintaining reconstruction fidelity while significantly reducing memory requirements.

Table 1: **CAVINR vs SOTA**. CAVINR shows better quality in reconstructing videos across datasets, as measured by PSNR and SSIM. 'F' refers to frame number, $\#\hat{\theta}'$ is the size of video-specific weights or video token size. Training time is measured in 'GPU hrs'.

| Methods | F | Encoder size | INR size↓ | $\#\hat{\theta}'$↓ | Epoch | GPU hrs↓ | PSNR↑ | | | | SSIM↑ | | | |
|---|---|---|---|---|---|---|---|---|---|---|---|---|---|---|
| | | | | | | | Train | K400 | SthV2 | UCF101 | Train | K400 | SthV2 | UCF101 |
| TransINR Chen & Wang (2022) | 4 | 48.0M | 99k | 25k | 150 | 63 | 23.7 | 22.1 | 24.6 | 22.1 | 0.659 | 0.631 | 0.728 | 0.622 |
| GINR Kim et al. (2022) | 4 | 47.6M | 139.4k | 25.6k | 150 | 65 | 24.5 | 23.2 | 25.9 | 23.1 | 0.685 | 0.66 | 0.744 | 0.66 |
| FastNeRV Chen et al. (2024) | 4 | 47.6M | 85.6k | 24.1k | 150 | 9 | 26.6 | 26.6 | 29.4 | 26 | 0.756 | 0.754 | 0.816 | 0.752 |
| CAVINR(ours) | 4 | 45.5M | 86.4K | 27k | 20 | 5 | _31.4_ | _31.5_ | _31.9_ | _31.5_ | _0.924_ | _0.922_ | _0.925_ | _0.923_ |
| CAVINR(ours) | 4 | 45.5M | 86.4K | 27k | 150 | 39 | **35.3** | **33.5** | **36.0** | **34.8** | **0.955** | **0.946** | **0.956** | **0.955** |
| TransINR Chen & Wang (2022) | 8 | 48.0M | 99k | 25k | 150 | 119 | 22.3 | 20.3 | 22.8 | 20.7 | 0.626 | 0.595 | 0.703 | 0.591 |
| GINR Kim et al. (2022) | 8 | 47.6M | 139.4k | 25.6k | 150 | 123 | 23.9 | 22.8 | 25.3 | 22.7 | 0.671 | 0.65 | 0.737 | 0.651 |
| FastNeRV Chen et al. (2024) | 8 | 47.6M | 85.6k | 24.1k | 150 | 11 | 25.8 | 25.8 | 28.5 | 25.2 | 0.732 | 0.727 | 0.795 | 0.723 |
| CAVINR(ours) | 8 | 46.3M | 86.4K | 27k | 20 | 9 | _28.8_ | _28.6_ | _28.9_ | _28.7_ | _0.896_ | _0.893_ | _0.897_ | _0.895_ |
| CAVINR(ours) | 8 | 46.3M | 86.4K | 27k | 150 | 70 | **33.1** | **31.1** | **30.5** | **29.8** | **0.939** | **0.927** | **0.916** | **0.913** |
| TransINR Chen & Wang (2022) | 16 | 48.0M | 99k | 25k | 150 | 234 | 21.5 | 18.4 | 21.1 | 19.2 | 0.615 | 0.555 | 0.678 | 0.561 |
| GINR Kim et al. (2022) | 16 | 47.6M | 139.4k | 25.6k | 150 | 242 | 21.7 | 21.7 | 24.2 | 21.7 | 0.647 | 0.624 | 0.72 | 0.625 |
| FastNeRV Chen et al. (2024) | 16 | 47.6M | 85.6k | 24.1k | 150 | 15 | 23.6 | 23.2 | 25.9 | 22.9 | 0.657 | 0.642 | 0.731 | 0.642 |
| CAVINR(ours) | 16 | 47.9M | 86.4K | 27k | 20 | 17 | _27.3_ | _27.1_ | _27.4_ | _27.3_ | _0.872_ | _0.869_ | _0.873_ | _0.872_ |
| CAVINR(ours) | 16 | 47.9M | 86.4K | 27k | 150 | 128 | **31.5** | **29.0** | **29.1** | **29.3** | **0.923** | **0.907** | **0.910** | **0.911** |

**Temperature-Modulated Attention Enhancement** While the coordinate-based cross-attention mechanism enables video token reconstruction, we observe that reconstruction fidelity can be further improved. Inspired by localized attention mechanisms Zhang et al. (2024), we enhance the decoder's representational capacity through temperature-modulated attention.

Traditional localized attention layers (LAL) implement threshold-based attention weight filtering:

$$\text{LAL} = \text{Norm}\left(\text{ReLU}\left(\text{Softmax}\left(\frac{\mathbf{Q}\mathbf{K}^\top}{\sqrt{d}}\right) - m\right)\right)\mathbf{V} \tag{11}$$

where threshold $m$ suppresses weak attention weights below a boundary. However, this recomputation introduces significant computational overhead and memory consumption.

To preserve the benefits of enhanced attention focus while improving computational efficiency, we introduce a temperature parameter $\tau$ to modulate attention distribution sharpness:

$$\mathbf{F} = \text{Softmax}\left(\frac{\mathbf{Q}\mathbf{K}^\top}{\tau \cdot \sqrt{d}}\right)\mathbf{V}, \quad \begin{cases} \mathbf{Q} = \phi(\gamma(\mathbf{x})) \\ \mathbf{K}, \mathbf{V} = \phi(\mathbf{T}_v) \end{cases} \tag{12}$$

By adjusting $\tau$, we control the concentration of attention across spatiotemporal locations: lower values sharpen attention distribution (similar to LAL's thresholding effect). CAVINR achieves comparable representational enhancement to LAL while maintaining computational efficiency through direct integration into the standard softmax operation.

## 4 EXPERIMENTS

### 4.1 EXPERIMENTAL SETUP

**Datasets.** Our evaluation employs three benchmark video datasets following the protocol established in Chen et al. (2024). Kinetics-400 (K400) Kay et al. (2017) serves as the primary training corpus, containing 240K videos spanning 400 action classes. For computational efficiency while maintaining class diversity, we utilize a curated subset of 10,000 videos (25 per class). Evaluation is performed on the test sets of K400, Something-Something V2 Goyal et al. (2017) (20K motion-centric videos) and UCF101 Soomro et al. (2012) (3.5K human-action videos).

**Implementation Details.** All experiments employ standardized video inputs at $256 \times 256$ resolution with temporal sampling of 4, 8, and 16 frames. The preprocessing pipeline consists of three stages: (1) aspect ratio preservation via shorter-side resizing to 256px, (2) center cropping for spatial alignment, and (3) uniform temporal sampling for consistency across sequences.

Our transformer hypernetwork processes $16 \times 16$ spatiotemporal patches through 6 encoder layers with hidden dimension 384, generating compact latent tokens for the decoding process. The coordinate-attentive decoder combines a single transformer block with a 2-layer MLP, mapping coordinate embeddings to YUV color values through SiLU-activated Elfwing et al. (2018) projections. Training employs the AdamW optimizer Loshchilov & Hutter (2017) with initial learning rate $10^{-4}$. All models are implemented in PyTorch Paszke et al. (2019) and trained on 8 NVIDIA A800 GPUs with Intel Xeon Gold 6430 CPUs @ 2.1GHz. Video fidelity is quantified through PSNR and SSIM metrics, computed frame-wise and averaged across temporal sequences.

Table 2: **CAVINR vs ANR Zhang et al. (2024)**. CAVINR shows better quality in reconstructing videos across datasets, as measured by PSNR and SSIM. Our method takes up less memory and trains faster, and can converge to a better performance.

| Methods | F | Token length $N$ | $\#\hat{\theta}'$ | Epoch | Memory per-batch ↓ | GPU hrs ↓ | PSNR ↑ Train | K400 | SthV2 | UCF101 | SSIM ↑ Train | K400 | SthV2 | UCF101 |
|---|---|---|---|---|---|---|---|---|---|---|---|---|---|---|
| ANR-S | 4 | 384 | 27k | 150 | 18G | 75 | 31.9 | 29.9 | 32.0 | 31.0 | 0.933 | 0.919 | 0.934 | 0.929 |
| CAVINR-S | 4 | 384 | 27k | 150 | 2.6G | 39 | **35.3** | **33.5** | **36.0** | **34.8** | **0.955** | **0.946** | **0.956** | **0.955** |
| ANR-M | 4 | 512 | 36k | 150 | 23G | 85 | 32.2 | 30.8 | 33.1 | 32.2 | 0.942 | 0.930 | 0.942 | 0.939 |
| CAVINR-M | 4 | 512 | 36k | 150 | 2.7G | 41 | **35.8** | **34.0** | **36.6** | **35.4** | **0.960** | **0.952** | **0.960** | **0.960** |
| ANR-L | 4 | 768 | 54k | 150 | 34G | 132 | 33.69 | 31.25 | 33.46 | 32.91 | 0.947 | 0.934 | 0.946 | 0.946 |
| CAVINR-L | 4 | 768 | 54k | 150 | 2.7G | 52 | **37.7** | **36.1** | **38.9** | **38.0** | **0.970** | **0.965** | **0.972** | **0.973** |
| ANR-S | 8 | 384 | 27k | 150 | 35G | 178 | 30.9 | 28.3 | 29.8 | 29.3 | 0.921 | 0.901 | 0.914 | 0.907 |
| CAVINR-S | 8 | 384 | 27k | 150 | 3.9G | 70 | **33.1** | **31.1** | **30.5** | **29.8** | **0.939** | **0.927** | **0.916** | **0.913** |
| ANR-M | 8 | 512 | 36k | 150 | 46G | 205 | 31.4 | 28.4 | 29.9 | 29.2 | 0.924 | 0.899 | 0.913 | 0.908 |
| CAVINR-M | 8 | 512 | 36k | 150 | 4G | 84 | **33.6** | **31.19** | **32.89** | **32.57** | **0.943** | **0.931** | **0.939** | **0.943** |
| ANR-L | 8 | 768 | 54k | 150 | 74G | 344 | 33.5 | 32.0 | 33.4 | 32.9 | 0.945 | 0.936 | 0.944 | 0.946 |
| CAVINR-L | 8 | 768 | 54k | 150 | 4G | 92 | **34.9** | **32.9** | **34.9** | **34.6** | **0.954** | **0.930** | **0.952** | **0.956** |

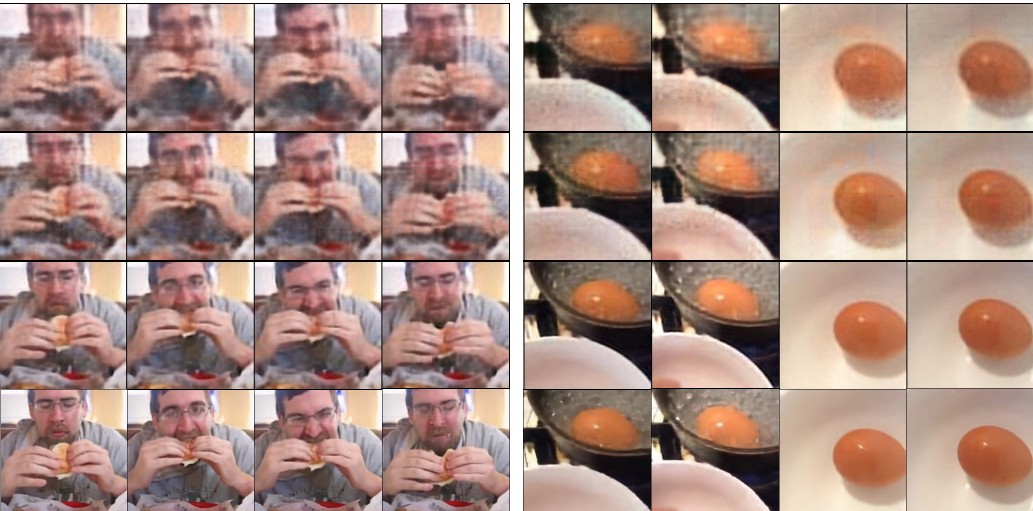

Figure 3: **Qualitative Comparison.** Visualizations for INR encoding methods: TransINR Chen & Wang (2022) (**Top line**), GINR Kim et al. (2023a) (**Second line**), FastNeRV Chen et al. (2024) (**Third line**), and CAVINR (**Bottom**, ours). Our method excels in reconstructing videos with superior fidelity and fine details. Best viewed digitally and zoomed in.

## 4.2 COMPARISON WITH STATE-OF-THE-ART METHODS

Table 1 presents comprehensive comparisons against leading INR methods including TransINR Chen & Wang (2022), GINR Kim et al. (2022), and FastNeRV Chen et al. (2024) across different frames (4, 8, and 16).

**Reconstruction Quality.** CAVINR demonstrates substantial improvements in reconstruction fidelity across all evaluation scenarios. At 4-frame resolution, our method achieves 35.3 dB PSNR on training data and maintains strong generalization with 33.5 dB on K400, 36.0 dB on SthV2, and 34.8 dB on UCF101, representing improvements of 8.7, 6.9, 6.6, and 8.8 dB respectively over the strongest baseline FastNeRV. Similar performance gains are observed across 8-frame and 16-frame configurations, with consistent SSIM improvements exceeding 0.1 across all datasets.

**Training Efficiency.** Our method demonstrates improved training efficiency, achieving better performance than baseline methods in just 20 epochs compared to their requirement of 150 epochs for convergence. This corresponds to a $7.5\times$ reduction in training time while maintaining competitive performance across all evaluation metrics. When CAVINR is trained for the full 150-epoch duration, the performance gains become more substantial, indicating both faster convergence and better final results compared to existing methods.

**Computational Cost.** CAVINR achieves superior reconstruction quality while maintaining computational efficiency comparable to existing methods. With 45.5–47.9M parameters, our model remains within the size range of current approaches while delivering improved performance. Train-

Table 3: **Ablation Study.** Evaluation of individual component contributions in CAVINR. Each component provides improvements in reconstruction quality and computational efficiency.

| RGB to YUV | Block Query | Convolution Tokenizer | Axis-Adaptive-Embedding | Temperature Transformer | Memory per-batch ↓ | GPU hrs ↓ | PSNR ↑ | | | | SSIM ↑ | | | |
|---|---|---|---|---|---|---|---|---|---|---|---|---|---|---|
| | | | | | | | Train | K400 | SthV2 | UCF | Train | K400 | SthV2 | UCF |
| | | | | | 35G | 178 | 26.9 | 25.3 | 26.3 | 26.2 | 0.668 | 0.579 | 0.612 | 0.602 |
| ✓ | | | | | 35G | 178 | 27.8 | 26.5 | 27.4 | 27.2 | 0.726 | 0.693 | 0.708 | 0.701 |
| | ✓ | ✓ | | | 18G | 178 | 27.8 | 26.5 | 27.4 | 27.2 | 0.726 | 0.693 | 0.708 | 0.701 |
| ✓ | ✓ | ✓ | | | 18G | 172 | 28.2 | 27.7 | 28.4 | 28.5 | 0.821 | 0.793 | 0.813 | 0.802 |
| ✓ | ✓ | | ✓ | | 18G | 180 | 28.8 | 27.8 | 28.5 | 28.8 | 0.825 | 0.804 | 0.816 | 0.812 |
| ✓ | ✓ | | | ✓ | 3.9G | 70 | 31.1 | 29.3 | 30.9 | 30.4 | 0.888 | 0.879 | 0.860 | 0.871 |
| ✓ | ✓ | ✓ | ✓ | ✓ | 3.9G | 70 | **33.1** | **31.1** | **32.8** | **32.3** | **0.939** | **0.927** | **0.935** | **0.938** |

ing efficiency is most apparent in the 4-frame configuration, where CAVINR converges in 39 GPU hours compared to FastNeRV's 150-epoch protocol.

**Visual Quality.** Figure 3 shows CAVINR's performance across different video sequences. Compared to TransINR, GINR, and FastNeRV, our method produces sharper details, preserves textures more effectively, and maintains better color accuracy. These improvements are particularly evident in sequences with rapid motion or fine textural patterns, where our approach generates cleaner edges and reduces visual artifacts.

## 4.3 MEMORY EFFICIENCY ANALYSIS

Table 2 presents a comprehensive comparison with the transformer-based method ANR Zhang et al. (2024), examining memory usage and computational efficiency across three model scales (Small, Medium, Large) and varying temporal resolutions. While ANR achieves competitive results in image reconstruction, for comprehensive evaluation, we implement video reconstruction experiments to compare its performance against our method.

**Memory Reduction.** The combination of block query processing and temperature-scaled attention mechanisms yields substantial memory efficiency improvements. CAVINR consistently maintains memory usage between 2.6–4.0 GB per batch across training configurations, while ANR requires 18–74 GB—corresponding to an 85–95% reduction in memory consumption. The memory advantage increases with model scale and temporal sequence length. For instance, ANR-L processing 8 frames demands 74 GB, whereas CAVINR only needs 4 GB for the same configuration.

**Training Speed.** The memory efficiency directly translates to accelerated training. CAVINR-L with 4 frames requires only 52 GPU hours compared to ANR-L's 132 hours, achieving 2.5× speedup while delivering superior reconstruction quality (37.7 vs 33.69 dB PSNR). This pattern holds consistently across all model configurations.

**Scalability.** The performance advantages of CAVINR become more pronounced with increased model capacity. CAVINR-L achieves the highest reconstruction quality while maintaining practical memory requirements, demonstrating excellent scalability properties that enable deployment of larger models within memory constraints.

## 4.4 ABLATION STUDY

Table 3 presents a comprehensive ablation study evaluating five key components in our CAVINR framework: RGB-to-YUV color space conversion, block query processing, convolutional tokenizer, spatiotemporal embedding, and temperature-modulated attention.

**Individual Component Analysis.** Starting from a baseline of 26.9 dB PSNR (35G memory, 178 GPU hours), RGB-to-YUV conversion delivers the first significant gain (+0.9 dB PSNR, +0.058 SSIM), validating perceptually-motivated color space representation. Block query processing maintains reconstruction quality while achieving 49% memory reduction (35G→18G). The convolutional tokenizer adds +0.4 dB PSNR and reduces training time to 172 hours, demonstrating superior learnable spatial abstraction over traditional patch methods.

**Spatiotemporal Enhancement.** Spatiotemporal embedding provides consistent improvements (+1 dB PSNR, +0.1 SSIM), confirming the value of explicit temporal modeling. Temperature-modulated attention yields the most dramatic gains: +2.3 dB PSNR with substantial efficiency improvements (memory: 3.9G, training time: 70 hours).

**Synergistic Effects.** The complete CAVINR framework achieves 33.1 dB PSNR and 0.939 SSIM—cumulative improvements of +6.2 dB and +0.271 SSIM over baseline. This shows strong component synergy that delivers both superior reconstruction quality and computational efficiency.

Table 4: **Video Compression Comparison.** Rate-distortion analysis comparing CAVINR with traditional codecs and neural compression methods (FastNeRV). CAVINR demonstrates favorable quality-size tradeoffs and flexible bit allocation while delivering competitive processing speeds.

| | AV1 | H.264 | | | FastNeRV | | | | | CAVINR (ours) | | | | |
| | CRF 60 | CRF 35 | CRF 40 | CRF 45 | 8 bits | 7 bits | 6 bits | 5 bits | 4 bits | 8 bits | 7 bits | 6 bits | 5 bits | 4 bits |
|---|---|---|---|---|---|---|---|---|---|---|---|---|---|---|
| Size(KB) ↓ | 21.9 | 20.4 | 13.1 | **8.7** | 23.7 | 20.7 | 17.7 | 14.7 | 11.6 | 26.9 | 23.6 | 20.2 | 16.9 | 13.5 |
| PSNR ↑ | 32.4 | 32.8 | 30.0 | 27.3 | 28.4 | 28.3 | 28.1 | 27.5 | 25.6 | 32.99 | 32.89 | **32.51** | 31.20 | 27.77 |
| SSIM ↑ | 0.910 | 0.912 | 0.860 | 0.788 | 0.808 | 0.807 | 0.784 | 0.712 | 0.802 | 0.938 | 0.937 | **0.933** | 0.919 | 0.866 |
| VPS ↑ | 313 | 447 | 460 | 485 | | | **5175** | | | | | 125 | | |

**Attention Mechanism Analysis.** Figure 4 shows how attention patterns evolve during training. Initially, diffuse patterns (a) sharpen into focused activations (b), while coordinate-specific patterns (c,d) reveal how different spatial locations activate distinct token subsets for efficient content-coordinate mapping. Comparing attention before and after training demonstrates the emergence of structured weights that correspond to relevant video content. This validates our temperature-modulated model learns to focus on pertinent information while suppressing irrelevant activations, achieving both spatial selectivity and semantic coherence.

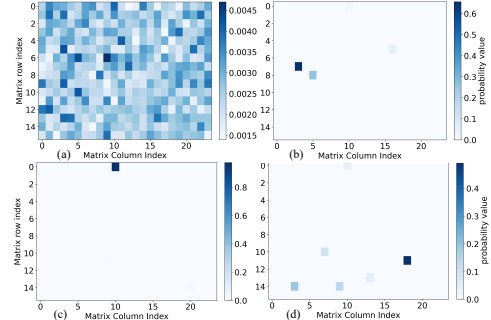

Figure 4: Token probability distribution visualizations.

**Temperature Coefficient Analysis.** We evaluate temperature coefficients $\tau$ ranging from 0.1 to 1.0 to optimize attention sharpness, measuring performance using PSNR and SSIM metrics as shown in Figure 5. Lower temperature values ($\tau = 0.1$) produce overly concentrated attention that can miss important contextual information, while higher values ($\tau = 1.0$) create diffuse patterns that reduce reconstruction precision. Our analysis shows that $\tau = 0.4$ provides the best balance, offering focused attention while maintaining adequate coverage.

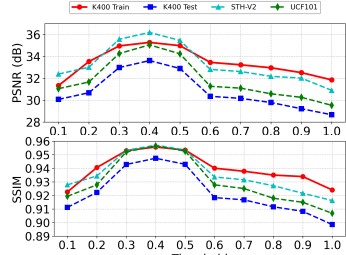

Figure 5: $\tau$–dependent results.

**Encoding Efficiency.** Figure 6 illustrates the substantial efficiency improvements achieved by our feed-forward approach over gradient-based optimization methods. CAVINR delivers encoding speeds $10^5\times$ faster than the NeRV baseline across multiple datasets, reducing video encoding from hour-long optimization procedures to millisecond-scale forward passes. This acceleration enables real-time deployment while preserving superior reconstruction fidelity. The encoding speedup results from eliminating iterative weight optimization by using persistent decoder parameters and pre-trained transformer hypernetworks.

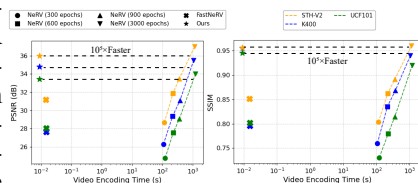

Figure 6: Encoding speed comparison across datasets.

### 4.5 DOWNSTREAM TASK: VIDEO COMPRESSION

**Performance Evaluation.** We evaluate CAVINR against traditional codecs (H.264, AV1) and neural methods (FastNeRV) using comprehensive rate-distortion analysis presented in Table 4. CAVINR significantly outperforms FastNeRV, achieving 32.51 dB PSNR and 0.933 SSIM at 6 bits (20.2KB) compared to FastNeRV's performance at 17.7KB. This represents substantial improvements of +4.4 dB PSNR and +0.131 SSIM, validating our coordinate-attentive decoding approach. Against traditional codecs, CAVINR's 6-bit configuration substantially outperforms H.264 CRF 40 (30.0 dB PSNR, 0.860 SSIM, 13.1KB) and matches H.264 CRF 35 quality (32.8 dB PSNR, 0.912 SSIM, 20.4KB) at comparable file sizes. Furthermore, CAVINR delivers superior quality with a smaller file size than AV1 CRF 60 (32.4 dB PSNR, 0.910 SSIM, 21.9KB). The flexible bit allocation enables diverse quality-size trade-offs, ranging from aggressive 4-bit compression (27.77 dB PSNR, 0.866 SSIM, 13.5KB) to high-quality 8-bit encoding (32.99 dB PSNR, 0.938 SSIM, 26.9KB), thus accommodating varied application requirements.

Table 5: **CAVINR vs ANR Zhang et al. (2024) with different resolution**. CAVINR shows better quality in reconstructing videos across datasets, as measured by PSNR and SSIM. Our method takes up less memory and trains faster, and can converge to a better performance.

| Methods | F | Frame Resolution | $\#\hat{\theta}'$ | Epoch | Memory per-batch ↓ | GPU hrs ↓ | PSNR ↑ | | | | SSIM ↑ | | | |
|---|---|---|---|---|---|---|---|---|---|---|---|---|---|---|
| | | | | | | | Train | K400 | SthV2 | UCF101 | Train | K400 | SthV2 | UCF101 |
| ANR-S | 4 | 512 | 27k | 50 | 64G | 275 | 27.9 | 27.2 | 28.0 | 27.0 | 0.833 | 0.819 | 0.834 | 0.829 |
| CAVINR-S | 4 | 512 | 27k | 50 | 5.2G | 62 | **29.3** | **29.5** | **29.0** | **29.8** | **0.855** | **0.846** | **0.856** | **0.855** |
| ANR-M | 4 | 512 | 36k | 50 | 76G | 285 | 28.2 | 27.8 | 28.1 | 28.2 | 0.842 | 0.830 | 0.842 | 0.839 |
| CAVINR-M | 4 | 512 | 36k | 50 | 5.3G | 66 | **29.8** | **29.7** | **29.6** | **29.9** | **0.860** | **0.852** | **0.860** | **0.860** |
| ANR-L | 4 | 512 | 54k | OOM | OOM | OOM | OOM | OOM | OOM | OOM | OOM | OOM | OOM | OOM |
| CAVINR-L | 4 | 512 | 54k | 50 | 5.3G | 70 | **30.7** | **30.1** | **30.9** | **30.5** | **0.870** | **0.865** | **0.872** | **0.873** |
| ANR-S | 1 | 1024 | 27k | 50 | 64G | 278 | 26.9 | 27.3 | 26.8 | 26.3 | 0.821 | 0.801 | 0.814 | 0.807 |
| CAVINR-S | 1 | 1024 | 27k | 50 | 5.2G | 64 | **29.1** | **29.1** | **28.5** | **28.8** | **0.839** | **0.827** | **0.816** | **0.813** |
| ANR-M | 1 | 1024 | 36k | 50 | 76G | 305 | 27.4 | 28.4 | 27.9 | 27.2 | 0.824 | 0.801 | 0.813 | 0.808 |
| CAVINR-M | 1 | 1024 | 36k | 50 | 5.3G | 68 | **29.6** | **28.2** | **29.9** | **29.7** | **0.843** | **0.831** | **0.839** | **0.843** |
| ANR-L | 1 | 1024 | 54k | OOM | OOM | OOM | OOM | OOM | OOM | OOM | OOM | OOM | OOM | OOM |
| CAVINR-L | 1 | 1024 | 54k | 50 | 5.3G | 71 | **30.2** | **29.5** | **30.4** | **30.2** | **0.854** | **0.840** | **0.852** | **0.856** |

**Processing Efficiency and Applications.** CAVINR processes 125 videos per second, representing a 41× reduction compared to FastNeRV's throughput of 5,175 VPS. This slower performance stems from the computational overhead introduced by our attention mechanisms. However, 125 VPS still far exceeds real-time decoding requirements for most practical use cases.

**Rate-Distortion Performance.** Figure 7 demonstrates CAVINR's competitive compression efficiency on HEVC class D datasets. Our method achieves PSNR ranging from 30.78 to 36.29 dB across varying bitrates. At 33 dB PSNR, CAVINR (0.062 bpp) outperforms the traditional HM-16.25 codec (0.069 bpp) while maintaining comparable quality to modern learned methods DCVC-FM and DCVC-RT. At higher quality levels around 35 dB, CAVINR continues to demonstrate efficient rate-distortion trade-offs, requiring 0.105 bpp compared to HM-16.25's 0.116 bpp. These results validate that our feed-forward approach preserves strong compression performance despite eliminating iterative optimization.

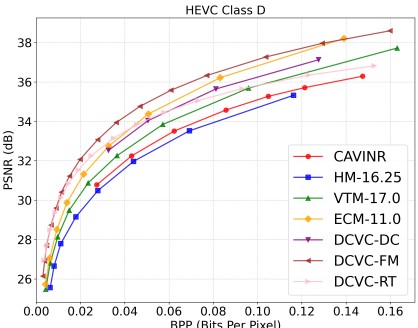

Figure 7: Rate-distortion comparison on HEVC class D sequences.

### 4.6 HIGH-RESOLUTION SCALABILITY ANALYSIS

We evaluate CAVINR's scalability at $512 \times 512$ and $1024 \times 1024$ resolutions, comparing against ANR Zhang et al. (2024). Table 5 summarizes the results.

**Resolution Scaling.** At 512 RES with 4 frames, CAVINR-S achieves 29.3 dB PSNR versus ANR-S's 27.9 dB. The gap increases with model capacity: CAVINR-L reaches 30.7 dB while ANR-L encounters out-of-memory (OOM) errors on 80GB A800 GPUs.

**Extreme Resolution.** At 1024 RES, CAVINR-S achieves 29.1 dB PSNR using 5.2 GB memory per batch, while ANR-S requires 64 GB—a 12× reduction. ANR-L fails entirely at this resolution, whereas CAVINR-L processes inputs successfully with 30.2 dB PSNR and 5.3 GB memory. These results validate that block query processing enables coordinate-based reconstruction at resolutions previously infeasible for transformer-based INR methods.

**Training Efficiency.** At 512 RES, CAVINR-S requires 62 GPU hours compared to ANR-S's 275 hours (4.4× speedup). This advantage persists at 1024 RES: CAVINR completes training in 64-71 GPU hours across all scales, while ANR demands 278-305 hours for smaller models and fails for larger configurations.

## 5 CONCLUSION

This paper presents CAVINR, a transformer-based framework that enhances video implicit neural representations through coordinate-aware cross-attention mechanisms. By replacing per-video optimization with shared transformer weights and video-specific tokens, our method achieves $10^5\times$ faster encoding than gradient-based NeRV while delivering 6–9 dB PSNR improvements over existing approaches. Beyond superior reconstruction quality, CAVINR provides substantial efficiency gains: 85–95% memory reduction compared to current techniques and 7.5× faster convergence with consistent performance across diverse video datasets. These advantages position CAVINR as a solution for large-scale video processing applications where both quality and efficiency are paramount.

## ETHICS STATEMENT

This work presents purely theoretical and computational research. The study does not involve any human subjects, human data collection, biological experiments, or interaction with living systems. All analyses are conducted on publicly available benchmark datasets or synthetic data, and no new datasets are released as part of this work. The methodologies and findings presented are of a fundamental nature and do not raise foreseeable risks of misuse, harmful applications, or significant societal impacts requiring specific ethical mitigation. There are no potential conflicts of interest, discriminatory biases, or privacy/security concerns directly arising from the research described in this paper. The authors confirm that this research adheres to the principles outlined in the ICLR Code of Ethics and standard academic integrity practices.

## REPRODUCIBILITY STATEMENT

To ensure reproducibility, we provide detailed implementation information in the appendix. The convolutional tokenization approach and axis-adaptive positional encoding scheme are described in full. We present the theoretical basis for our cross-attention mechanism and temperature-modulated attention optimization. All hyperparameter settings and training procedures are documented, including optimizer configurations and learning rate schedules. We provide quantitative computational complexity analysis compared to baseline methods. The data processing pipelines and standardized preprocessing steps are described for all benchmark datasets. Code will be released upon publication.

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

# A APPENDIX

## A.1 VIDEO TOKENIZATION METHODS

We compare two paradigms for converting video data into spatiotemporal tokens: the conventional unfold-based method and our proposed convolutional tokenizer. Given an input video tensor $\mathbf{V} \in \mathbb{R}^{T \times C \times H \times W}$, both methods partition frames into patches but diverge fundamentally in projection strategies. This tokenization approach builds upon the success of patch-based representations in vision transformers Dosovitskiy et al. (2020) and extends these concepts to video domains Arnab et al. (2021).

**Baseline Unfold Tokenizer.** The baseline unfold tokenizer first merges batch and temporal dimensions ($\mathbf{V}' \in \mathbb{R}^{T \times C \times H \times W}$), then extracts patches via unfold operations with patch size $P \times P$ Dosovitskiy et al. (2020):

$$\mathbf{V}_{\text{unfold}} = \texttt{unfold}(\mathbf{V}') \in \mathbb{R}^{T \times (C \cdot P^2) \times L} \tag{13}$$

where $L$ denotes patches per frame. Complex reshaping operations follow: temporal separation, permutation, and flattening to $\mathbf{V}_{\text{flat}} \in \mathbb{R}^{(L \cdot T) \times (C \cdot P^2)}$. Finally, a linear layer projects patches to $D$-dimensional tokens:

$$\mathbf{Z} = \texttt{Linear}(C \cdot P^2 \to D)(\mathbf{V}_{\text{flat}}) \tag{14}$$

This method incurs high computational overhead from explicit patch-vector construction and intermediate tensor manipulations.

**Proposed Convolutional Tokenizer.** Our proposed convolutional tokenizer replaces unfold operations with learnable convolutional projections:

$$\mathbf{V}_{\text{conv}} = \texttt{Conv2d}(C \to D, \text{kernel\_size} = P, \text{stride} = P)(\mathbf{V}') \tag{15}$$

where $\mathbf{V}_{\text{conv}} \in \mathbb{R}^{T \times D \times H' \times W'}$ and $H' = \lfloor H/P \rfloor$, $W' = \lfloor W/P \rfloor$. We then flatten spatial dimensions and restore temporal structure:

$$\mathbf{Z} = \texttt{reshape}(\texttt{flatten}(\mathbf{V}_{\text{conv}})) \in \mathbb{R}^{(T \cdot H' \cdot W') \times D} \tag{16}$$

This approach provides several key advantages: computational efficiency through eliminating costly reshaping operations via unified patch extraction and projection, enhanced learnability by enabling adaptive optimization of spatial feature extraction compared to static unfold operations Liu et al. (2021), and implementation simplicity by reducing tokenization to three streamlined operations while avoiding error-prone permutations. While both methods generate $T \cdot (H/P) \cdot (W/P)$ tokens, our approach better aligns with modern architectures that leverage convolutional inductive biases for vision tasks LeCun et al. (1998); Krizhevsky et al. (2012).

## A.2 AXIS-ADAPTIVE POSITIONAL ENCODING

Our Axis-Adaptive Positional Encoding (AAPE) addresses a fundamental limitation in existing coordinate encoding methods: the assumption that spatial and temporal dimensions require identical frequency characteristics. Video data exhibits distinct spectral properties across different dimensions—spatial information concentrates in higher frequencies for texture preservation, while temporal information resides primarily in lower frequencies due to motion coherence constraints.

**Frequency Scaling Parameter Selection.** The frequency scaling parameter $\sigma$ selection follows from the Nyquist-Shannon sampling theorem applied to natural video content Mallat (1999). For video resolution $H \times W \times T$, we determine the optimal spatial and temporal frequency parameter as:

$$\sigma_{\text{spatial}} = 2 \times \max(H, W), \sigma_{\text{temporal}} = 2 \times T. \tag{17}$$

This relationship ensures that positional encoding captures spatial frequencies up to the effective bandwidth of the video representation. When $\sigma$ equals the video resolution, the encoding approaches critical sampling and may introduce aliasing artifacts. Large values ($\sigma_s \geq 1024$) lead to high-frequency dominance, which reduces sensitivity to coarse spatial structures—a known issue in harmonic analysis Tancik et al. (2020); Mildenhall et al. (2020). Based on empirical evaluation, $\sigma_s = 512$ offers a good balance between fine-detail discrimination and stable coarse-scale representation for $256 \times 256$ videos.

**Adaptive Frequency Allocation.** We differentially allocate frequency components based on information density. Spatial dimensions receive increased allocation:

$$k_s = \lfloor 4k/3 \rfloor \tag{18}$$

enabling capture of fine details, edges, and textures critical for video quality. Temporal dimensions receive reduced allocation:

$$k_t = \lfloor k/3 \rfloor \tag{19}$$

reflecting lower temporal resolution and different coherence characteristics compared to spatial dimensions.

**Spectral Stratification Principle.** Natural images exhibit power-law spectral decay with most information in low-to-medium frequencies Field (1987); Ruderman (1994). Our spatial encoding with $\sigma^{i/(k_s-1)}$ provides logarithmic frequency spacing matching this distribution. Video temporal dynamics operate at lower frequencies due to frame rate limitations and motion coherence Adelson & Bergen (1985). Our temporal encoding provides attenuated progression suitable for temporal pattern capture.

The frequency allocation strategy follows information-theoretic principles. Spatial dimensions exhibit high mutual information with visual content due to rich spatial structure, while temporal dimensions show lower mutual information due to redundancy and coherence. Our encoding distributes available representational capacity ($6k$ dimensions) according to information content, following rate-distortion optimization principles Cover & Thomas (2006).

A.3 CROSS-ATTENTION CONVERGENCE ANALYSIS

We establish theoretical justification for cross-attention superiority over dynamic weight generation in video implicit neural representation. Under Lipschitz continuity assumptions Arjovsky et al. (2017), cross-attention converges faster than dynamic weight methods. Let $f_\theta(x)$ denote traditional dynamic weight generation Sitzmann et al. (2020) and $g_\phi(T_v, x)$ denote our cross-attention mechanism with video tokens $T_v$ and coordinate queries $x$.

Dynamic weight generation updates as:

$$\theta^{(t+1)} = \theta^{(t)} - \eta \nabla_\theta L(f_\theta(x), y) \tag{20}$$

Cross-attention updates as:

$$\phi^{(t+1)} = \phi^{(t)} - \eta \nabla_\phi L(g_\phi(T_v, x), y) \tag{21}$$

The cross-attention formulation maintains persistent weights $\phi$ while enabling content-adaptive reconstruction through attention mechanisms. This leads to more stable optimization landscapes Dosovitskiy et al. (2020) and provides implicit regularization preventing overfitting to specific video instances Morerio et al. (2017). The cross-attention operation:

$$\text{Attention}(Q, K, V) = \text{softmax}\left(\frac{QK^T}{\sqrt{d}}\right) V \tag{22}$$

inherently normalizes influence across video tokens, providing regularization effects absent in direct weight modulation approaches Vaswani et al. (2017)–Bahdanau et al. (2015).

A.4 TEMPERATURE-MODULATED ATTENTION

We derive the optimal temperature parameter $\tau$ through information-theoretic analysis Hinton et al. (2015)–Jang et al. (2017). Temperature-modulated attention controls attention distribution entropy Shannon (1948):

$$H(\tau) = -\sum_i p_i(\tau) \log p_i(\tau) \tag{23}$$

where $p_i(\tau) = \frac{\exp(a_i/\tau)}{\sum_j \exp(a_j/\tau)}$. The optimal temperature balances attention focus and coverage:

$$\tau^* = \arg\min_\tau \mathcal{L}_{recon} + \lambda \cdot \mathcal{R}_{entropy}(\tau) \tag{24}$$

where $\mathcal{R}_{entropy}(\tau) = |H(\tau) - H_{target}|$ penalizes deviations from target entropy Pereyra et al. (2017). Empirical validation across multiple datasets identifies $\tau^* \approx 0.4$ as optimal, achieving superior reconstruction quality while maintaining computational efficiency Müller et al. (2019).

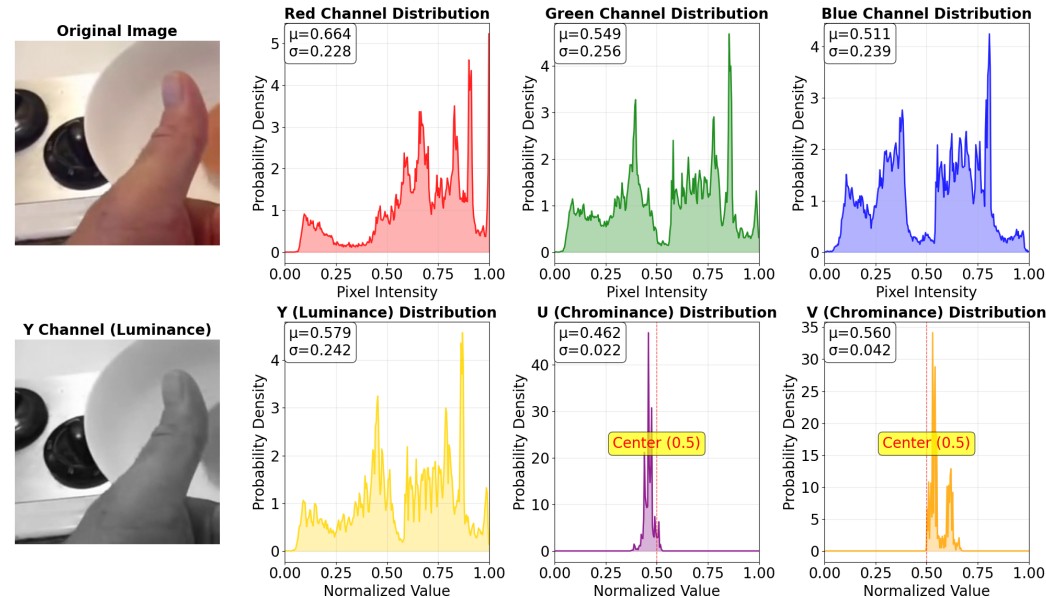

Figure 8: **RGB vs YUV** RGB channels show discrete distributions across the full range $[0, 1]$, while YUV $U\&V$ channels concentrate around $0.5$, enabling efficient compression.

### A.5 COLOR SPACE OPTIMIZATION

We adopt the YUV color space instead of RGB to exploit the statistical properties of natural video content for efficient tokenization. The RGB-to-YUV transformation follows ITU-R BT.601 standard ITU-R (2011):

$$
\begin{aligned}
Y &= 0.299 \cdot R + 0.587 \cdot G + 0.114 \cdot B \\
U &= -0.169 \cdot R - 0.331 \cdot G + 0.500 \cdot B + 128 \\
V &= 0.500 \cdot R - 0.419 \cdot G - 0.081 \cdot B + 128
\end{aligned}
\tag{25}
$$

where $Y$ represents luminance information, while $U$ and $V$ encode chrominance information.

Unlike RGB channels with discrete distributions across the full intensity range Gonzalez & Woods (2017), YUV demonstrates asymmetric statistical properties. The luminance channel $Y$ maintains broad distributions capturing structural information Poynton (2012), while chrominance channels $U$ and $V$ concentrate around neutral values (0.5 normalized) Salomon (2004) as shown in Figure 8. This asymmetry enables adaptive compression where fewer token dimensions encode chrominance due to limited dynamic range, while luminance receives primary representational capacity.

Our YUV-based tokenization achieves information density optimization by prioritizing the most informative luminance channel, perceptual alignment with human visual sensitivity favoring luminance over chrominance Wandell (1995), and computational efficiency through reduced chrominance dimensionality. We implement channel-wise adaptive allocation where luminance utilizes $\alpha \cdot D$ dimensions ($\alpha = 0.6$) while chrominance channels share the remaining $(1 - \alpha) \cdot D$ dimensions equally, achieving superior compression ratios while preserving visual quality compared to RGB approaches Le Gall & Tabatabai (2020).

### A.6 DETAILED TRAINING CONFIGURATION

As detailed in Table 6 (Complete Hyperparameter Settings), our CAVINR model implementation incorporates carefully tuned architectural and training parameters across multiple components.

**CAVINR Architecture:** The model uses a token dimension of 72 with a token length of 384, implemented with six encoder layers, each containing six attention heads with 64-dimensional heads. The multi-layer perceptron (MLP) components have a hidden dimension of 768 and a feed-forward

Table 6: Complete Hyperparameter Settings

| Parameter | Value |
|---|---|
| **CAVINR Architecture** | |
| Token dimension (d) | 72 |
| Token length | 384 |
| Number of encoder layers (L) | 6 |
| Number of attention heads | 6 |
| Head dimension | 64 |
| MLP hidden dimension | 768 |
| Feed-forward dimension | 3072 |
| Sigma parameter ($\sigma$) | 256 |
| Number of groups (n_groups) | 64 |
| Block query size (M) | 64 |
| Temperature parameter ($\tau$) | 0.4 |
| **Transformer Encoder Parameters** | |
| Input dimension | 3 |
| Output dimension | 3 |
| Output bias | 0.5 |
| Network depth | 2 |
| Hidden dimension | 72 |
| PE dimension | 24 |
| PE sigma | 512 |
| Number of frames | 4 |
| Activation function | SiLU |
| Rescale | False |
| **Training Parameters** | |
| Optimizer | AdamW |
| Learning rate | 1e-4 |
| Weight decay | 1e-5 |
| Batch size | 2 |
| Total epochs | 150 |
| Warmup epochs | 10 |
| LR schedule | Cosine annealing |
| **Data Processing** | |
| Input resolution | 256×256 |
| Frame sampling | Uniform |
| Color space | YUV |
| Normalization | [0, 1] |

dimension of 3072. Key parameters include a sigma parameter $\sigma = 256$, $n\_groups = 64$, block query size $M = 64$, and temperature parameter $\tau = 0.4$.

**Transformer Encoder Configuration:** The transformer encoder processes 3-dimensional input and output with an output bias of 0.5. The network consists of 2 layers with a hidden dimension of 72. Positional encoding uses PE dimension 24 and PE sigma 512, processing 4 frames with Swish activation and no rescaling.

**Training Setup:** We use the AdamW optimizer with a learning rate of $1 \times 10^{-4}$ and weight decay of $1 \times 10^{-5}$. Training uses a batch size of 2 for 150 epochs, with 10 warmup epochs followed by cosine annealing learning rate scheduling for stable convergence.

**Data Processing:** Input videos are processed at $256 \times 256$ resolution with uniform frame sampling. Frames are converted to the YUV color space and normalized to $[0, 1]$ for consistent input scaling across the dataset.

Table 7: Computational complexity comparison for single frame reconstruction. FLOPs (Floating Point Operations) and MACs (Multiply-Accumulate Operations) are measured in billions (G) for standard video resolution.

| Method | FLOPs (G) | MACs (G) | Architecture Type |
|---|---|---|---|
| NeRV | 1.30 | 0.65 | Convolutional |
| ANR | 239.73 | 119.45 | Transformer-based |
| CAVINR (Ours) | 31.92 | 15.95 | Temperature Attention |

## A.7 COMPUTATIONAL COMPLEXITY ANALYSIS

We conduct a comprehensive computational complexity analysis comparing CAVINR with baseline approaches. Table 7 presents computational requirements for reconstructing a single video frame across different methods.

The analysis shows key trade-offs between architectural approaches in neural video representation. NeRV achieves high efficiency at 1.30 GFLOPs per frame through direct coordinate-to-pixel mapping using learned convolutional layers, avoiding attention mechanism overhead entirely. However, this efficiency limits global dependency modeling and reduces reconstruction quality for complex video content. ANR requires significantly higher computation at 239.73 GFLOPs—approximately $184\times$ more than NeRV. This overhead results from the transformer architecture's need to recalculate attention weights for each spatial-temporal coordinate during reconstruction. The quadratic complexity of self-attention creates computational bottlenecks that become especially problematic at high resolutions, where coordinate queries scale quadratically with spatial dimensions. CAVINR strikes a middle ground at 31.92 GFLOPs, reducing computational complexity by approximately $7.5\times$ compared to ANR while maintaining attention-based representational capabilities. This efficiency gain comes from our temperature-modulated attention mechanism, which reduces computation through simplified attention weight calculation, efficient spatial-temporal token interactions enabled by axis-adaptive positional encoding, and optimized attention patterns that lower memory bandwidth requirements during forward passes.

**Memory Scaling Properties.** Memory complexity patterns differ substantially across methods. Traditional NeRV approaches scale as $O(N_{\text{params}} \times N_{\text{videos}})$ for weight storage, requiring individual networks per video sequence. This scaling becomes prohibitive for large video collections. CAVINR achieves $O(N_{\text{tokens}} \times d + N_{\text{params}}^{\text{fixed}})$ complexity through shared decoder weights, significantly reducing memory requirements for multi-video scenarios where $N_{\text{params}}^{\text{fixed}} \ll N_{\text{params}} \times N_{\text{videos}}$.

**Forward Pass Analysis.** NeRV requires $O(H \times W \times T \times N_{\text{params}})$ operations for coordinate-wise convolutions, while CAVINR needs $O(H \times W \times T \times N_{\text{tokens}} + N_{\text{tokens}} \times d^2)$ for cross-attention operations. Despite the quadratic term in token interactions, reduced token count and optimized attention computation yield practical efficiency gains, particularly for high-resolution videos where $N_{\text{tokens}} \ll H \times W \times T$.

**Training Efficiency.** Per-video optimization requires $O(E \times B \times N_{\text{params}})$ where $E$ represents epochs and $B$ represents batch size. CAVINR achieves $O(B \times N_{\text{shared}})$ where $N_{\text{shared}} \ll N_{\text{params}}$, enabling more efficient multi-video training. This shared parameter approach reduces training time by $7.5\times$ compared to gradient-based optimization methods while achieving superior reconstruction quality.

**Scalability Considerations.** Our block query processing technique addresses the quadratic memory bottleneck inherent in attention mechanisms. By restricting coordinate queries to local $M \times M$ windows, we reduce memory complexity from $O((HWT)^2)$ to $O(M^2 \times \frac{HWT}{M^2}) = O(HWT)$, enabling practical processing of high-resolution, long-sequence videos within reasonable computational constraints while preserving reconstruction quality.

While CAVINR cannot match the raw efficiency of purely convolutional approaches like NeRV, it provides compelling trade-offs by achieving substantial computational savings over transformer-based methods while delivering superior reconstruction quality. This efficiency enables practical deployment of attention-based neural video representation in resource-constrained environments where both quality and computational efficiency are critical.

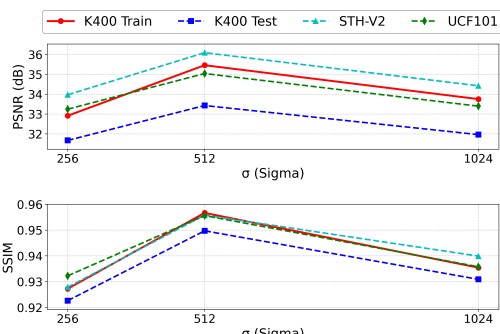

Figure 9: **Sigma Parameter.** PSNR/SSIM performance across training and test sets at varying $\sigma$: Insufficient dimensionality ($\sigma = 256$) limits representation quality, excessive dimensionality ($\sigma = 1024$) introduces noise, while $\sigma = 512$ achieves optimal capacity-representation balance.

### A.8 HYPERPARAMETER SENSITIVITY ANALYSIS

To validate our architectural design choices, we analyze four key hyperparameters that affect CAV-INR performance: sigma dimension, encoder depth, MLP depth, and batch size.

**Sigma Parameter Analysis.** Figure 9 shows how feature dimension $\sigma$ affects reconstruction quality through an inverted-U relationship. Low dimensionality ($\sigma = 256$) limits representational capacity, degrading PSNR and SSIM across datasets. High dimensionality ($\sigma = 1024$) introduces excessive parameters that cause overfitting and noise amplification, particularly evident in test set degradation. The optimal value ($\sigma = 512$) balances representational capacity with generalization, achieving peak performance across K400 Train, K400 Test, STH-V2, and UCF101.

**Encoder Depth Optimization.** Figure 10 shows monotonic improvement with encoder depth $L$. Shallow architectures ($L = 6$) provide insufficient hierarchical feature extraction, while progressive increases to $L = 8$ and $L = 10$ yield consistent gains through enhanced multi-scale representations. Peak performance at $L = 12$ reflects improved feature hierarchy construction, with gains most apparent in complex temporal datasets like STH-V2.

**MLP Depth Impact.** Figure 11 shows a positive relationship between MLP depth and performance. Shallow networks (depth=2) have limited non-linear transformation capacity, which restricts complex video-to-coordinate mapping. Moderate depths (3-4 layers) provide gradual improvements, while depth=5 achieves the best expressiveness, effectively capturing complex spatial-coordinate relationships.

**Batch Size Sensitivity.** Figure 12 reveals inverse correlation between batch size and performance. Small batches (size=2) enable precise gradient estimation crucial for coordinate-based learning, while larger batches progressively degrade performance through gradient over-smoothing that reduces optimization dynamics. Batch size 16 performs the poorest, confirming that coordinate-based neural video representations require small-batch training.

The analysis indicates that our architectural choices are appropriate, with consistent patterns across different video datasets.

### A.9 CROSS-DOMAIN RECONSTRUCTION

We assess CAVINR's reconstruction through zero-shot inference across diverse video domains using models trained exclusively on K400 data. The evaluation encompasses medical imaging datasets—EchoCP Wang et al. (2021) (30 patients with PFO diagnosis videos) and EchoNet-LVH Duffy et al. (2022) (12,000 parasternal-long-axis echocardiography videos).

#### A.9.1 QUALITATIVE ANALYSIS

Figure 13 shows a visual comparison of reconstruction quality on echocardiographic sequences. The results highlight differences in each method's ability to handle medical imaging challenges, including high noise levels and complex anatomical structures.

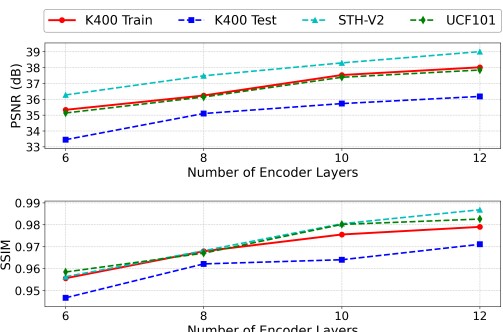

Figure 10: **Encoder Depth.** PSNR/SSIM performance across training and test sets at varying layer depth $L$: Shallow architecture ($L = 6$) limits hierarchical feature extraction, intermediate depths ($L = 8, 10$) show progressive improvements, while $L = 12$ achieves optimal performance through deeper feature hierarchies.

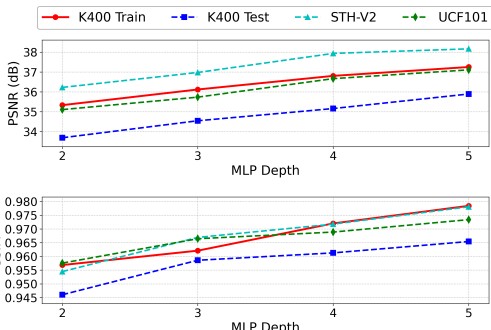

Figure 11: **MLP Depth.** PSNR/SSIM performance across training and test sets at varying MLP depths: Shallow networks (Depth = 2) limit non-linear transformations, moderate depths (Depth = 3, 4) enhance feature mapping complexity, while Depth = 5 achieves maximum expressiveness.

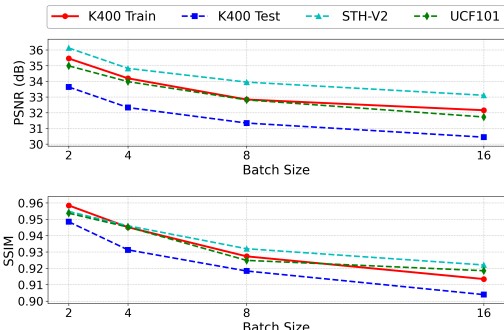

Figure 12: **Batch Size.** PSNR/SSIM performance across training and test sets at varying batch sizes: Large batches (Size = 16) cause over-smoothed gradients, limiting optimization, medium batches (Size = 4, 8) reduce gradient noise at convergence cost, while Size = 2 achieves optimal gradient estimation precision.

FastNeRV produces blurred outputs that obscure important anatomical details, struggling with cardiac wall boundaries and tissue textures essential for clinical assessment. The method exhibits significant loss of high-frequency information, reducing the diagnostic clarity needed for medical applications. ANR shows improvement with better reconstruction of anatomical features and improved handling of ultrasound noise characteristics. However, fine details and subtle tissue variations remain poorly resolved, particularly in regions with complex echogenicity patterns.

CAVINR achieves better reconstruction quality, preserving fine anatomical details while maintaining consistent performance across varying noise conditions. The method effectively balances noise suppression with detail preservation, producing reconstructions that closely match ground truth quality. CAVINR's advantages are particularly apparent in challenging low signal-to-noise regions where other methods struggle to maintain adequate image quality.

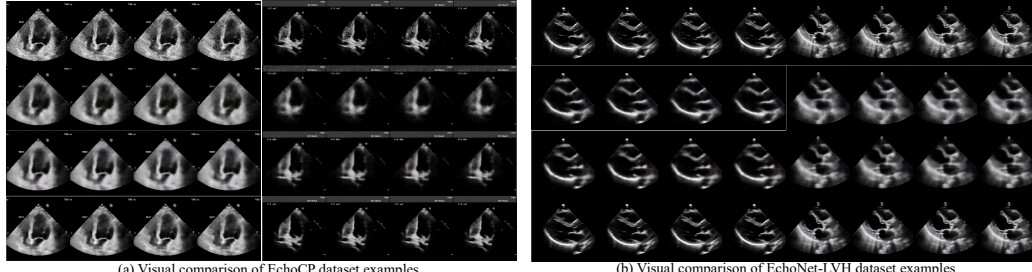

(a) Visual comparison of EchoCP dataset examples    (b) Visual comparison of EchoNet-LVH dataset examples

Figure 13: **Qualitative Comparison.** Visualizations for Cross-Domain reconstruction: Ground Truth (**Top line**), FastNeRV Chen et al. (2024) (**Second line**), ANR Zhang et al. (2024) (**Third line**), and CAVINR (**Bottom**, ours). Our method excels in reconstructing medical videos with fine details. Best viewed digitally and zoomed in.

Table 8: Comparative Performance of Video Reconstruction Methods on Medical Imaging Datasets

| Dataset | Method | PSNR (dB) ↑ | SSIM ↑ | PSNR Gain | SSIM Gain |
|---------|--------|-------------|--------|-----------|-----------|
| EchoCP | FastNeRV | 20.53 | 0.607 | - | - |
|  | ANR | 24.40 | 0.784 | +3.87 | +0.177 |
|  | **CAVINR** | **26.85** | **0.864** | **+6.32** | **+0.257** |
| EchoNet-LVH | FastNeRV | 22.88 | 0.684 | - | - |
|  | ANR | 27.45 | 0.835 | +4.57 | +0.151 |
|  | **CAVINR** | **29.64** | **0.916** | **+6.76** | **+0.232** |

### A.9.2 QUANTITATIVE RESULTS

Table 8 presents a comprehensive quantitative evaluation of the three reconstruction methods across two medical imaging datasets. The results demonstrate CAVINR's substantial superiority over existing approaches in both objective metrics.

On the EchoCP dataset, CAVINR achieves a PSNR of 26.85 dB and SSIM of 0.864, representing improvements of 6.32 dB and 0.257 respectively compared to FastNeRV, and 2.45 dB and 0.080 improvements over ANR. For the larger EchoNet-LVH dataset, CAVINR attains even more pronounced gains with 29.64 dB PSNR and 0.916 SSIM, corresponding to 6.76 dB and 0.232 improvements over FastNeRV, and 2.19 dB and 0.081 improvements over ANR.

CAVINR consistently outperforms FastNeRV across both datasets, achieving average gains of 6.54 dB in PSNR and 0.245 in SSIM. These significant improvements demonstrate its efficacy in overcoming key challenges specific to medical video reconstruction—notably enhancing noise resilience while preserving diagnostically relevant image features.

### A.10 COMPARISON WITH NEURAL VIDEO COMPRESSION

We compare encoding and decoding speeds against neural video codecs DCVC-DC Li et al. (2023), DCVC-FM Li et al. (2024), and DCVC-RT Jia et al. (2025). Table 9 reports speeds in frames per second (fps) across multiple GPUs.

**Encoding Speed.** CAVINR achieves the fastest encoding across all configurations. On 1080p videos, CAVINR encodes at 247.0 fps on A100, outperforming DCVC-RT (125.2 fps, 1.97×) and DCVC-DC (3.3 fps, 74.8×). This advantage results from feed-forward processing that avoids iterative optimization and complex entropy coding pipelines.

**Decoding Speed.** CAVINR decodes at 7.7 fps on A100 for 1080p, lower than DCVC-RT's 112.8 fps. This difference reflects architectural trade-offs: DCVC-RT uses optimized convolutional decoders for parallel frame generation, while CAVINR performs cross-attention for pixel-level reconstruction. Nevertheless, 7.7 fps exceeds requirements for video archival and offline processing applications.

**Hardware and Resolution Scaling.** Performance scales consistently across GPUs from datacenter (A100) to consumer hardware (RTX 2080 Ti). At 720p, CAVINR achieves 12.7 fps decoding on A100 (1.65× over 1080p), demonstrating sub-linear complexity growth.

Table 9: **Speed analysis**. The encoding / decoding speed (measured in frames per second, fps) are evaluated across various resolutions and devices, including the NVIDIA A100, NVIDIA A6000, RTX 4090, and RTX 2080 Ti.

| Model | A100 | A6000 | 4090 | 2080Ti |
|---|---|---|---|---|
| DCVC-DC | 3.3 / 4.3 | 1.7 / 2.2 | 2.3 / 2.9 | 0.8 / 1.4 |
| DCVC-FM | 5.0 / 5.9 | 3.1 / 3.8 | 3.7 / 4.4 | 1.9 / 2.3 |
| DCVC-RT | 125.2 / 112.8 | 70.4 / 63.8 | 118.8 / 105.3 | 39.5 / 34.1 |
| CAVINR | 247.0 / 7.7 | 168.4 / 6.5 | 236.8 / 7.4 | 94.6 / 3.4 |

(a) Coding speed on $1920 \times 1080$ videos.

| Model | A100 | A6000 | 4090 | 2080Ti |
|---|---|---|---|---|
| DCVC-DC | 6.5 / 7.9 | 3.5 / 4.3 | 5.5 / 6.7 | 2.1 / 2.9 |
| DCVC-FM | 8.5 / 9.4 | 5.9 / 6.6 | 9.3 / 10.4 | 4.0 / 4.7 |
| DCVC-RT | 173.9 / 149.2 | 147.3 / 132.5 | 225.1 / 185.2 | 73.3 / 67.0 |
| CAVINR | 250.0 / 12.7 | 202.3 / 8.5 | 240.6 / 12.4 | 98.5 / 5.4 |

(b) Coding speed on $1280 \times 720$ videos.

Table 10: **Zero-shot super-resolution comparison between CAVINR and ANR Zhang et al. (2024).** Models are trained on single frames at $128 \times 128$ resolution and evaluated at both native resolution and $2\times$ super-resolution ($256 \times 256$) without additional training. CAVINR achieves superior generalization with +2.2 dB PSNR improvement at $2\times$ upscaling while requiring $12\times$ less training time. SR denotes super-resolution inference.

| Methods | Frame Resolution | $\#\hat{\theta}'$ | Epoch | GPU hrs ↓ | PSNR ↑ Train | K400 | SthV2 | UCF101 | SSIM ↑ Train | K400 | SthV2 | UCF101 |
|---|---|---|---|---|---|---|---|---|---|---|---|---|
| ANR | 128 | 27k | 150 | 96 | 33.1 | 33.2 | 33.2 | 33.6 | 0.933 | 0.919 | 0.934 | 0.929 |
| CAVINR | 128 | 27k | 50 | 8 | **33.3** | **33.3** | **33.4** | **33.8** | **0.935** | **0.926** | **0.936** | **0.935** |
| ANR (SR 2×) | 256 | 27k | 0 | 0 | 28.9 | 29.3 | 28.8 | 28.3 | 0.861 | 0.841 | 0.874 | 0.867 |
| CAVINR (SR 2×) | 256 | 27k | 0 | 0 | **31.1** | **30.1** | **31.5** | **31.8** | **0.919** | **0.907** | **0.916** | **0.913** |

## A.11 SUPER-RESOLUTION VIA VARIATIONAL COORDINATES

To demonstrate CAVINR's downstream task capabilities, we conduct zero-shot super-resolution experiments following the variational coordinate approach from ANR Zhang et al. (2024). Both models are trained on single frames at $128 \times 128$ resolution and directly evaluated at $256 \times 256$ ($2\times$ upscaling) without additional training, by querying coordinates on a denser grid.

As shown in Table 10, CAVINR achieves comparable quality at native resolution while requiring only 8 GPU hours versus ANR's 96 hours ($12\times$ speedup). For $2\times$ super-resolution, CAVINR demonstrates significantly better generalization with **+2.2 dB PSNR improvement** on average, validating that our coordinate-attentive design enables practical downstream applications unavailable to autoencoder-based methods.

