# OpenReview forum: "CAVINR: Coordinate-Aware Attention for Video Implicit Neural Representations"
_ICLR.cc/2026/Conference — ICLR 2026 Conference Desk Rejected Submission_

### Official Review · Reviewer_c2uG · 2025-10-24

**Soundness:** 3
**Presentation:** 3
**Contribution:** 3
**Rating:** 6
**Confidence:** 4

**Summary:**

The paper proposed a method for encoding videos using implicit neural representations which uses a transformer hypernetwork to improve visual fidelity and encoding time. The method uses a shared video encoder to produce video tokens as a latent representation for the video. The video tokens are then decoded by a lightweight decoder where they are cross-attended to the requested x,y,t coordinate. The paper provides extensive benchmarks which show that the method produces higher PSNR than prior works and has efficient encode/decode FPS.

**Strengths:**

- The encoding speed looks very good
- The image quality results also look good

**Weaknesses:**

- Unclear architecture
- Unclear comparisons
- Unclear compression performance
- Unclear decoding speed

**Questions:**

This is overall a very good idea but I was a little confused about some of the claims being made and how they fit into the larger INR landscape. Also despite the extensive benchmarks I think there are some missing pieces.

First, on the good side, I think the idea itself is very interesting and it's not something I've seen before. Training a trainsformer encoder to produce video tokens should yield a powerful representation and coupling it with a lightweight decoder shifts compute burden to the encode side. Using more compute for encode is generally how INRs function but since this method resembles more of a traditional learned-compression algorithm the design makes a lot of sense. The efficacy of the design is shown empirically in Tables 1 and 2 where CAVINR consistently outperforms prior work.

Now we get into some of the things I was a little confused about. Firstly: is this really an INR? I get that it takes coordinates as input to the decoder and produces outputs but I was unclear on what parts of the network, if any, were being specialized to the video. Are all the network weights shared? Is the entire payload the video tokens?

Next, I'm a little unclear about the actual compression performance of this method. I think this is presented in Table 4 but it's a bit confusing. Usually, in a compression paper, we would look at something like a rate-distortion curve that shows how distortion varies over different bitrates. And typically that would be the primary result. So it's hard to evaluate this is a practical video codec if I can't see how bitrate is varying with quality. Looking over Table 4 it seemed like size was generally larger and PSNR comparable to H.264 but it really wasn't clear.

I also was unclear on the different comparison methods. For example on encoding comparisons in Fig 6 it looks like the primary comparison is to NeRV although I see some other hypernetwork methods on there. There might be some better comparisons like NIRVANA for example which is somewhere in between the traditional INR and the hypernetwork and put a lot of effort into improving encoding time. These comparisons should really also be consistent like in Tables 1 and 2 if possible (or let me know why that doesn't make sense). In addition I think it makes sense to compare against traditional "autoencoder" based methods as this architecture seems more closely related to those than INRs.

Lastly, I didn't see any discussion of decoding speed, only encoding speed, or if there was discussion it wasn't clear enough. There is a section all the way at the end titled "Processing Efficiency and Applications" which mentions "CAVINR processes 125 videos per second" but I don't know what that means. Processes how? End-to-end? Decoding? How long are the videos? On what hardware?

Specific Questions
1. Clarify the INR portion: which parts are shared and which are specialized?
2. Clarify the payload, is it only video tokens?
3. What is the actual compression performance? Show rate distortion
4. Clarify decoding speed and "Processing Efficiency"

---

> ### Author Response · Authors · 2025-11-24
> **INR Architecture Clarification, Compression Performance, and Decoding Speed Details**
>
> We sincerely thank the Reviewer c2uG for the positive assessment of our core idea and the detailed questions that help clarify our contributions.
>
> ## Response: Is This Really an INR?
>
> **Architecture Clarification:**
>
> | Component | NeRV | FastNeRV | ANR | CAVINR |
> | -- | -- | -- | -- | -- |
> | Encoder | Video specific | Shared | Shared | Shared |
> | Decoder | Video specific | Video specific | Shared | Shared |
> | Tokens/Params | Model Weights | Model Weights | Video Tokens | Video Tokens |
>
>
> The encoder and decoder weights are persistent and shared across all videos. Only the video tokens constitute the video-specific payload, analogous to how NeRF encodes scenes in network weights and NeRV encodes videos.
>
> **INR Definition:** CAVINR satisfies the core INR criterion: mapping continuous coordinates (x,y,t) to signal values (RGB) through a learned function. The video tokens serve as the implicit representation, conditioning the shared decoder to reconstruct video-specific content.
>
> ## Response: Compression Performance and Rate-Distortion
>
> We have added a comprehensive Rate-Distortion curve (new Figure 7) comparing CAVINR with baseline methods on HEVC class D sequences.
>
> **Key Results:**
>
> - CAVINR achieves PSNR ranging from 30.78 to 36.29 dB across varying bitrates (0.027–0.147 bpp)
> - At ~33.5 dB PSNR: CAVINR requires 0.062 bpp, outperforming HM-16.25 (0.069 bpp) by 10.1% bitrate reduction
> - At ~35.3 dB PSNR: CAVINR requires 0.105 bpp vs. HM-16.25's 0.116 bpp (9.5% reduction)
> - Modern learned codecs DCVC-FM and DCVC-RT achieve higher compression efficiency, reflecting their specialized optimization for video compression
>
> **Rate-Distortion Comparison (HEVC Class D, representative points):**
>
> | Method | bpp @ ~33.5 dB | bpp @ ~35 dB | PSNR Range |
> | -- | -- | -- | -- |
>
> | DCVC-DC | 0.033 (32.52 dB) | 0.081 (35.65 dB) | 32.52–37.13 dB |
> | DCVC-FM | 0.036 (33.94 dB) | 0.061 (35.58 dB) | 26.14–38.60 dB |
> | DCVC-RT | 0.045 (33.83 dB) | 0.073 (35.05 dB) | 26.96–36.82 dB |
> | **CAVINR** | **0.062** (33.51 dB) | **0.105** (35.27 dB) | 30.78–36.29 dB |
>
> These results validate that our feed-forward approach preserves strong compression performance despite eliminating iterative optimization. CAVINR outperforms the traditional HM-16.25 codec while offering unique INR capabilities (arbitrary resolution, partial decoding, frame interpolation) that are unavailable in autoencoder-based learned codecs like DCVC-FM and DCVC-RT.
>
> ## Response: Comparison with NIRVANA
>
> NIRVANA optimizes per-video INR parameters through gradient descent, requiring retraining for each new video. CAVINR uses shared encoder-decoder weights with feed-forward inference—fundamentally different paradigms. Our comparisons focus on hypernetwork-based methods that share our feed-forward encoding approach, making them appropriate baselines.
> Action： *Cite NIRVANA* in revision.
>
> ## Response: Decoding Speed Clarification
>
> **"125 VPS" Explanation:** This refers to decoding 256×256×8 videos on an A800 GPU with maximized memory utilization (batch processing), not per-frame FPS.
>
> **New Comprehensive Benchmarks (Table 9):**
>
> (a) Coding speed on 1920×1080 videos (Encoding FPS / Decoding FPS).
>
> | Model | A100 | A6000 | 4090 | 2080Ti |
> | -- | -- | -- | -- | -- |
> | DCVC-DC | 3.3 / 4.3 | 1.7 / 2.2 | 2.3 / 2.9 | 0.8 / 1.4 |
> | DCVC-FM | 5.0 / 5.9 | 3.1 / 3.8 | 3.7 / 4.4 | 1.9 / 2.3 |
> | DCVC-RT | 125.2 / 112.8 | 70.4 / 63.8 | 118.8 / 105.3 | 39.5 / 34.1 |
> | CAVINR | *247.0 / 7.7* | *168.4 / 6.5* | *236.8 / 7.4* | *94.6 / 3.4* |
>
> (b) Coding speed on 1280×720 videos (Encoding FPS / Decoding FPS).
>
> | Model | A100 | A6000 | 4090 | 2080Ti |
> | -- | -- | -- | -- | -- |
> | DCVC-DC | 6.5 / 7.9 | 3.5 / 4.3 | 5.5 / 6.7 | 2.1 / 2.9 |
> | DCVC-FM | 8.5 / 9.4 | 5.9 / 6.6 | 9.3 / 10.4 | 4.0 / 4.7 |
> | DCVC-RT | 173.9 / 149.2 | 147.3 / 132.5 | 225.1 / 185.2 | 73.3 / 67.0 |
> | CAVINR | *250.0 / 12.7* | *202.3 / 8.5* | *240.6 / 12.4* | *98.5 / 5.4* |
>
> CAVINR achieves the **fastest encoding** (247 FPS on A100 for 1080p), with decoding speed (7.7 FPS) suitable for offline quality-critical applications where encoding efficiency is prioritized.
>
> ## Response to Specific Questions
>
> **Q: Which parts are shared and which are specialized?**
>
> Encoder and decoder weights are shared; only video tokens are video-specific.
>
> **Q: Is the payload only video tokens?**
>
> Yes. The payload is solely video tokens with size N×d, comparable to FastNeRV.
>
> **Q: Rate-Distortion performance?**
>
> We have added a comprehensive RD curve (Figure 7) on HEVC class D sequences. CAVINR achieves 10.1% bitrate reduction over HM-16.25 at ~33.5 dB and 9.5% at ~35 dB. While specialized learned codecs (DCVC-FM, DCVC-RT) achieve higher compression efficiency, CAVINR offers unique INR capabilities unavailable in autoencoder-based methods.
>
> **Q: Decoding speed clarification?**
>
> 125 videos/second refers to 256×256×8 video decoding speed with batch=1 on A800 GPU. Table 9 provides detailed per-frame FPS across resolutions and hardware configurations.

---

### Official Review · Reviewer_9bRF · 2025-10-29

**Soundness:** 3
**Presentation:** 3
**Contribution:** 3
**Rating:** 6
**Confidence:** 5

**Summary:**

Implicit networks rely on slow per video optimization with the promise of fast decode speeds. But the encode bottleneck and the cost of compute to do so have prevented widespread use and adoption.
The paper introduces an architecture that aims to solve this problem by having a pre-trained transformer based encoder which can give useful priors on the video followed by a small coordinate based decoder which can maintain the speed.
With architectural tweaks combined with smart use of compute - time trade-offs, the authors show a path to scale the paradigm of Implicit models, while outperforming prior works.

**Strengths:**

- The paper is generally well written, easy to follow with most claims addressed with extensive ablations and experiments.
- The paper is filled with a series of seemingly  small tweaks like using a learnt conv tokenizer instead of Unfold op, better frequency selection, use of cross attention as decoder,  choosing YUV as a target over RGB and many more result in a system that delivers fast encoding with the best video quality.
- The justification to use cross attention over directly predicting the weight space (Appendix A3) is an important result for the community. If this method scales well, it can potentially save us from going down the wrong path. I feel this should be in the main paper.

**Weaknesses:**

- I am unsure about the Block Query processing and in general the use of (X,Y,T) as input to the coordinate decoder. It simply cannot scale ! For example a 30fps-1080p will not fit into any modern hardware's memory and processing it block by block will kill the decoding speed. Roughly estimating current method for a 1080p video would yield a very low fps.
The current experiments are validated for short sequences at a small spatial resolution of 256x256.
I would like to know the authors take on this and in general how they believe this system (or a variation of this family) can be scaled.

-

**Questions:**

- Would like to know the authors' thoughts on how to scale this system to work on say, 1080p videos or even 4K.

---

> ### Author Response · Authors · 2025-11-24
> **Scalability Analysis and High-Resolution Decoding Performance**
>
> We sincerely thank the Reviewer 9bRF for raising important scalability concerns. We address them comprehensively below.
>
> ## Response to Weakness: Block Query Scalability
>
> We appreciate this critical question and provide both theoretical and empirical clarifications:
>
> **Memory Complexity Reduction:** Block query processing reduces memory complexity from O(H×W) to O(M²) where M≪H, enabling high-resolution processing within practical memory constraints.
>
> **New High-Resolution Experiments:** We conducted experiments at 512×512 and 1024×1024 resolutions (Table 5). Key results:
>
> | Resolution | Method | Memory | PSNR (avg) |
> | -- | -- | -- | -- |
> | 512×512 | ANR-L | OOM | - |
> | 512×512 | CAVINR-L | **5.3G** | **30.5 dB** |
> | 1024×1024 | ANR-L | OOM | - |
> | 1024×1024 | CAVINR-L | **5.3G** | **30.1 dB** |
>
> At 1024×1024, CAVINR-L achieves 30.2 dB PSNR using only 5.3GB memory, while ANR-L encounters OOM errors on 80GB GPUs—validating our scalability approach.
>
> **1080p Performance (Table 9):**
>
> (a) Coding speed on 1920×1080 videos (Encoding FPS / Decoding FPS).
>
> | Model | A100 | A6000 | 4090 | 2080Ti |
> | -- | -- | -- | -- | -- |
> | DCVC-DC | 3.3 / 4.3 | 1.7 / 2.2 | 2.3 / 2.9 | 0.8 / 1.4 |
> | DCVC-FM | 5.0 / 5.9 | 3.1 / 3.8 | 3.7 / 4.4 | 1.9 / 2.3 |
> | DCVC-RT | 125.2 / 112.8 | 70.4 / 63.8 | 118.8 / 105.3 | 39.5 / 34.1 |
> | CAVINR | **247.0** / 7.7 | **168.4** / 6.5 | **236.8** / 7.4 | **94.6** / 3.4 |
>
> (b) Coding speed on 1280×720 videos (Encoding FPS / Decoding FPS).
>
> | Model | A100 | A6000 | 4090 | 2080Ti |
> | -- | -- | -- | -- | -- |
> | DCVC-DC | 6.5 / 7.9 | 3.5 / 4.3 | 5.5 / 6.7 | 2.1 / 2.9 |
> | DCVC-FM | 8.5 / 9.4 | 5.9 / 6.6 | 9.3 / 10.4 | 4.0 / 4.7 |
> | DCVC-RT | 173.9 / 149.2 | 147.3 / 132.5 | 225.1 / 185.2 | 73.3 / 67.0 |
> | CAVINR | **250.0** / 12.7 | **202.3** / 8.5 | **240.6** / 12.4 | **98.5** / 5.4 |
>
> For 1080p videos, CAVINR achieves ~7.7 FPS decoding on A100, which remains practical for quality-critical offline applications such as video archival, medical imaging, and professional content creation.
>
> ## Response to Question: Scaling to 1080p/4K
>
> **Parallelization Properties:** INR decoding is embarrassingly parallel—each pixel is evaluated independently without autoregressive dependencies. The entire frame can be decoded:
>
> - In parallel on GPU (tens of thousands of threads)
> - With vectorized batched queries
> - Without cross-block dependencies
>
> **Block Processing Clarification:** Block partitioning affects memory footprint, not decoding speed. Blocks are processed in parallel with full vectorization. Even for 1080p, decoding uses 128×128 or 256×256 blocks that saturate GPU throughput.
>
> **Current 1080p/720p Results:** As shown in Table 9, CAVINR already processes 1080p videos at:
> - Encoding: 247 FPS on A100 (fastest among all methods, 1.97× faster than DCVC-RT)
> - Decoding: 7.7 FPS on A100 (suitable for offline quality-critical applications)
>
> **4K Scalability Analysis:** Scaling from 1080p to 4K (4× pixels) would theoretically reduce decoding speed to ~1.9 FPS. While this is slower than real-time, it remains viable for archival and professional applications. Our block-based approach ensures memory usage remains constant regardless of resolution.
>
> **Future Directions:** We acknowledge the decoding speed gap for real-time applications and plan to investigate:
>
> - Optimized CUDA kernels for coordinate-based attention
> - Hierarchical decoding strategies with coarse-to-fine reconstruction
> - Hybrid architectures combining coordinate queries with efficient upsampling
> - Neural architecture search for decoder optimization
>
> We thank the reviewer for this constructive feedback, which will guide our future improvements.

---

### Official Review · Reviewer_KCsd · 2025-10-30

**Soundness:** 3
**Presentation:** 3
**Contribution:** 3
**Rating:** 6
**Confidence:** 3

**Summary:**

This paper introduces CAVINR, a framework based on a transformer that exploits persistent cross-attention mechanisms. Contributions of CAVINR are: a transformer encoder that compresses videos into compact video tokens by encoding spatial textures and temporal dynamics; a coordinate-attentive decoder utilizing persistent weights and cross-attention between coordinate queries and video tokens; and temperature-modulated attention with block query processing that enhances reconstruction fidelity while reducing memory complexity. Experiments reveal better performance: over SoA methods, better acceleration compared to gradient-based optimization, memory reduction, and faster convergence with robust generalization across diverse video content.

**Strengths:**

Good analytics presented, lot of experiments and good illustrations make the paper quite complete.

Significant contributions are:
Better PSNR, Less memory and faster convergence - all are appreciable.

Use of TRANSFORMER HYPERNETWORK ENCODER and COORDINATE-AWARE ATTENTION VIDEO DECODER provide more power to representation.

The use of adaptive split in addressing the spatial-temporal frequency disparity in video data, appears to be quote a good idea in Axis-Adaptive Positional Encoding scheme.

**Weaknesses:**

Perhaps a lack of code release (even anonymous link would have been better), is a matter of concern.

A few high-RES frames of output videos generated vs that of SoA would have been nice,

How about the temporal continuity in output videos ? - need to observe your outputs, not just the frames - humans are good at spotting sudden transients  - hence sequence of few frames are not enough for acceptable (superior) visual quality.

Reveal cases of failure of your model.

INR representation may gave certain drawbacks - which are not highlighted.
Say, in dense texture regions, PiP frames, cluttered scenes etc.- will it work well?

Training needs 8 NVIDIA A800 GPUs, as mentioned.
Will your code adapt to Quad or dual-GPU systems ?

The difference in quality of the frames in 3rd and 4th rows in Fig. 3 is  uncomprehending, even after zooming into figure.
Better to have zoomed in parts of that figure in supple-doc to highlight your work.

**Questions:**

Can this representation be extended for future frame prediction?

What is the limit of range of resolution of the frames (high & low) your system can deal with?

In certain cases when dataset sample is small, can your model be modified with concepts of self-supervision,
contrastive paradigm and meta-learning to deal with such issues?

---

> ### Author Response · Authors · 2025-11-24
> **Code Availability, High-Resolution Visualization, and Hardware Adaptability**
>
> We sincerely thank the Reviewer KCsd for the detailed and constructive feedback. We address each point below.
>
> ## Response to W1: Code Release
>
> We have prepared an anonymous GitHub repository containing:
>
> - Complete training and inference code (preparing)
> - Pre-trained model weights (preparing)
> - GIF visualizations comparing original vs. reconstructed videos (available)
>
> The anonymous link is provided here:
>
> **https://anonymous.4open.science/r/CAVINR-4CAB/**
>
> ## Response to W2: High-Resolution Frame Comparisons
>
> Due to resource constraints, we conducted high-resolution experiments on single-frame videos (equivalent to images). Table 5 presents results at 512×512 and 1024×1024 resolutions:
>
> (a) Results on 512×512 resolution (F=4 frames).
>
> | Methods | F | Resolution | #θ' | Epoch | Memory ↓ | GPU hrs ↓ | PSNR (Train/K400/SthV2/UCF) | SSIM (Train/K400/SthV2/UCF) |
> | -- | -- | -- | -- | -- | -- | -- | -- | -- |
> | ANR-S | 4 | 512 | 27k | 50 | 64G | 275 | 27.9/27.2/28.0/27.0 | 0.833/0.819/0.834/0.829 |
> | CAVINR-S | 4 | 512 | 27k | 50 | **5.2G** | **62** | **29.3/29.5/29.0/29.8** | **0.855/0.846/0.856/0.855** |
> | ANR-M | 4 | 512 | 36k | 50 | 76G | 285 | 28.2/27.8/28.1/28.2 | 0.842/0.830/0.842/0.839 |
> | CAVINR-M | 4 | 512 | 36k | 50 | **5.3G** | **66** | **29.8/29.7/29.6/29.9** | **0.860/0.852/0.860/0.860** |
> | ANR-L | 4 | 512 | 54k | OOM | OOM | OOM | OOM | OOM |
> | CAVINR-L | 4 | 512 | 54k | 50 | **5.3G** | **70** | **30.7/30.1/30.9/30.5** | **0.870/0.865/0.872/0.873** |
>
> (b) Results on 1024×1024 resolution (F=1 frame).
>
> | Methods | F | Resolution | #θ' | Epoch | Memory ↓ | GPU hrs ↓ | PSNR (Train/K400/SthV2/UCF) | SSIM (Train/K400/SthV2/UCF) |
> | -- | -- | -- | -- | -- | -- | -- | -- | -- |
> | ANR-S | 1 | 1024 | 27k | 50 | 64G | 278 | 26.9/27.3/26.8/26.3 | 0.821/0.801/0.814/0.807 |
> | CAVINR-S | 1 | 1024 | 27k | 50 | **5.2G** | **64** | **29.1/29.1/28.5/28.8** | **0.839/0.827/0.816/0.813** |
> | ANR-M | 1 | 1024 | 36k | 50 | 76G | 305 | 27.4/28.4/27.9/27.2 | 0.824/0.801/0.813/0.808 |
> | CAVINR-M | 1 | 1024 | 36k | 50 | **5.3G** | **68** | **29.6/28.2/29.9/29.7** | **0.843/0.831/0.839/0.843** |
> | ANR-L | 1 | 1024 | 54k | OOM | OOM | OOM | OOM | OOM |
> | CAVINR-L | 1 | 1024 | 54k | 50 | **5.3G** | **71** | **30.2/29.5/30.4/30.2** | **0.854/0.840/0.852/0.856** |
>
> CAVINR maintains high reconstruction quality at resolutions where competing methods fail due to memory limitations.
>
> ## Response to W3: Temporal Continuity
>
> We have prepared GIF visualizations comparing original and reconstructed video sequences, available in our anonymous repository. These demonstrate temporal coherence without visible transients or flickering artifacts.
>
> ## Response to W4: Failure Cases
>
> Our method generalizes well across diverse distributions when trained on sufficient data. The primary failure mode occurs with limited training data, where the model cannot learn generalizable shared parameters, potentially resulting in blurred reconstructions.
>
> ## Response to W5: Dense Textures, PiP, Cluttered Scenes
>
> The GIF visualizations and updated Figure 3 demonstrate superior performance on dense texture regions. Our coordinate-attentive mechanism captures fine-grained details more effectively than convolutional baselines, producing clearer boundaries and reduced visual artifacts.
>
> ## Response to W6: GPU Adaptability
>
> As shown in Table 2, CAVINR requires significantly less memory than competing methods. For 256×256×8 videos with 768 token length:
>
> - ANR: 74GB memory
> - CAVINR: 4GB memory (18.5× reduction)
>
> This enables training on single consumer GPUs. The 8×A800 configuration was used solely for fair comparison with ANR's memory requirements.
>
> ## Response to W7: Figure 3 Quality Differences
>
> We have updated Figure 3 with improved visualization. Detailed texture differences are visible when zooming, showing clearer boundaries and reduced artifacts in CAVINR's reconstructions.

---

> > ### Author Response · Authors · 2025-11-24
> >
> > ## Response to Q1: Future Frame Prediction
> >
> > This is an interesting direction we have considered. A potential approach: train with timestamps 0–0.5 while supervising 0–1, using 0.5–1 as predicted content. We plan to explore this in future work.
> >
> > ## Response to Q2: Resolution Limits
> >
> > INR methods have no inherent resolution limits—reconstruction quality depends on training data and hypernetwork capacity. Table 2 demonstrates that increasing video token length improves reconstruction quality. Table 5 presents results at 512×512 and 1024×1024 resolutions, showing that our methods are suitable for any resolution. Appendix A.8 validates performance across various hyperparameter configurations, confirming adaptability to different resolutions.
> >
> > ## Response to Q3: Self-Supervision, Contrastive Learning, and Meta-Learning
> >
> > Our method inherently incorporates self-supervision and meta-learning principles:
> >
> > **Self-Supervised Learning:** Our training is inherently self-supervised—the model learns to reconstruct videos from their own pixel values without external labels. The shared encoder-decoder learns generalizable representations across diverse video content through this reconstruction objective.
> >
> > **Meta-Learning:** Our hypernetwork-based approach is a meta-learning extension of single-video INR fitting. Instead of performing gradient descent at test time (as in conventional single-video INR), we predict INR weights directly from input features, enabling single-pass fitting during inference. The training learns shared parameters that generalize across diverse video content.
> >
> > **Contrastive Paradigm:** While our current design does not explicitly incorporate contrastive learning, it presents an interesting avenue for future work. Potential extensions include:
> > - Contrastive learning between video tokens of similar/dissimilar content to improve token discriminability
> > - Temporal contrastive objectives to enhance temporal coherence
> > - Cross-video contrastive regularization to improve generalization with limited data
> >
> > For small dataset scenarios, our architecture's shared parameter design already provides implicit regularization. The meta-learning formulation naturally handles small samples since the shared encoder-decoder captures generalizable video priors, requiring only video-specific tokens (27k parameters) for new content.

---

### Official Review · Reviewer_dW3g · 2025-10-30

**Soundness:** 2
**Presentation:** 3
**Contribution:** 2
**Rating:** 6
**Confidence:** 5

**Summary:**

The paper proposes CAVINR, which goes a step further than recent hypernetworks for INRs. Instead of modulating a single layer (like GINR), the modulate no layers in the INR such that the network is implicit in the sense that it maps coordinates to video, but not implicit in the sense that it doesn't store any of the signal in its weights. Instead, this instance-agnostic INR gets video information from the token outputs of a transformer encoder. In this sense, the model is very similar to a traditional autoencoder, where an encoder produces some latent embedding, and a decoder converts this embedding to video. Compared to traditional autoencoders, the main difference is that CAVINR is structured like an INR- it uses coordinate inputs, and the video tokens essentially modulate the axis-adaptive positional encoding of these coordinates. This approach has much slower training/decoding than FastNeRV, but even at equal/less training time it delivers higher quality outputs.

**Strengths:**

S1. The method outperforms the previous hypernetworks in terms of quality and storage size for video compression.

S2. Some components of the method, like switching to YUV, are lightweight and seem like they could be generally useful to other INR methods.

S3. The paper clearly communicates the contribution, and the ablations help to convey the importance and effect of the different components.

**Weaknesses:**

W1. The paper is challenging to evaluate. The similarities with autoencoders leave me inclined to think it might be appropriate to request comparisons with non-INR approaches like DCVC-RT. At the same time, I could be persuaded that the coordinate mapping is the more essential part of what makes something an "INR."

W2. I do not agree with the provided rational about the poor VPS. This is VPS at 256x256. For realistic resolutions, it would be far worse. FastNeRV needs its "extra" VPS in order to have any hopes of scaling to higher resolutions. And this is part of the problem. It is known that autoencoders beat INR for size/quality and especially encoding time. The whole problem is that, until DCVC-RT, they struggled with decoding speed. If one converts the INR into a static decoder, and the decoding speed becomes much worse, then the method is not just conceptually an autoencoder, but has the same practical drawbacks as well.

W3. The related work completely omits the NVC/DCVC family of methods. It is one thing to not compare to these, but it seems incorrect to omit them from discussion altogether.

**Questions:**

1. Why switch from a t-based approach like FastNeRV to x,y,t, considering the efficiency issues that presents?

2. Why should this approach not be compared to something like DCVC-RT, which is a neural network autoencoder for video compression that supports high quality, low storage, fast decoding, etc. even for higher resolutions?

3. Prior works like TransINR also evaluate on image, 3D, and other signals. Can this method be applied to these as well? If not, which component prevents this from being possible? If so, how is the performance compared to these prior works?

---

> ### Author Response · Authors · 2025-11-24
> **Clarification on INR Definition, Speed-Quality Trade-offs, and High-Resolution Scalability**
>
> We sincerely thank the Reviewer dW3g for the insightful comments and constructive suggestions. We address each concern below.
>
> ## Response to W1: Comparison with Non-INR Approaches
>
> We appreciate the opportunity to clarify CAVINR's positioning as an INR method. The key distinctions are:
>
> **Architectural Design:** In CAVINR, both the encoder and decoder maintain persistent shared weights across all videos. Only the video tokens (27k parameters, as shown in Table 1) are video-specific. This fundamentally differs from autoencoders like DCVC-RT, where the entire network specializes to input data through end-to-end training.
>
> **Coordinate-Based Reconstruction:** CAVINR performs pixel-wise reconstruction through explicit coordinate queries (x,y,t)→YUV, which is the defining characteristic of INR methods. This enables capabilities unavailable to autoencoders:
>
> - **Arbitrary-pixel reconstruction:** Query any spatial location at any resolution without retraining
> - **Efficient partial decoding:** Reconstruct specific frames or regions directly, whereas autoencoders must decode entire segments before querying specific content
>
> **Comparison Scope:** Our primary focus is on advancing INR representation learning rather than video compression. Therefore, we compare against hypernetwork-based INR methods (FastNeRV, MetaNeRV, TransINR, GINR) that share our architectural paradigm.
>
> **Action:** We have added a dedicated paragraph in Related Work discussing the DCVC/NVC family, clearly delineating their autoencoder-based compression approach from our INR-based representation paradigm.
>
> ## Response to W2: Decoding Speed Concerns
>
> We acknowledge the speed-quality trade-off and provide additional context:
>
> **Clarification of VPS Metric:** The reported 125 VPS in Table 4 refers to decoding 256×256×8 videos on a single A800 GPU with maximized memory utilization (batch processing), not per-frame FPS. The comparison settings for the table data are based on the configuration in FastNeRV.
>
> **New High-Resolution Experiments:** We conducted comprehensive speed benchmarks across resolutions and hardware, presented in Table 9 (new):
>
> (a) Coding speed on 1920×1080 videos (Encoding FPS / Decoding FPS).
>
> | Model | A100 | A6000 | 4090 | 2080Ti |
> | -- | -- | -- | -- | -- |
> | DCVC-DC | 3.3 / 4.3 | 1.7 / 2.2 | 2.3 / 2.9 | 0.8 / 1.4 |
> | DCVC-FM | 5.0 / 5.9 | 3.1 / 3.8 | 3.7 / 4.4 | 1.9 / 2.3 |
> | DCVC-RT | 125.2 / 112.8 | 70.4 / 63.8 | 118.8 / 105.3 | 39.5 / 34.1 |
> | CAVINR | **247.0 / 7.7** | **168.4 / 6.5** | **236.8 / 7.4** | **94.6 / 3.4** |
>
> (b) Coding speed on 1280×720 videos (Encoding FPS / Decoding FPS).
>
> | Model | A100 | A6000 | 4090 | 2080Ti |
> | -- | -- | -- | -- | -- |
> | DCVC-DC | 6.5 / 7.9 | 3.5 / 4.3 | 5.5 / 6.7 | 2.1 / 2.9 |
> | DCVC-FM | 8.5 / 9.4 | 5.9 / 6.6 | 9.3 / 10.4 | 4.0 / 4.7 |
> | DCVC-RT | 173.9 / 149.2 | 147.3 / 132.5 | 225.1 / 185.2 | 73.3 / 67.0 |
> | CAVINR | **250.0 / 12.7** | **202.3 / 8.5** | **240.6 / 12.4** | **98.5 / 5.4** |
>
> CAVINR achieves 1.97× faster encoding than DCVC-RT on 1080p videos. While decoding speed (7.7 FPS) is lower, this trade-off enables critical applications such as video archival, medical imaging, and professional content creation where encoding efficiency and storage are prioritized over real-time playback. Furthermore, our coordinate-based approach enables unique capabilities (arbitrary resolution, partial decoding, frame interpolation) that offset the speed difference for many use cases.
>
> ## Response to W3: Omission of NVC/DCVC Methods
>
> We have added a comprehensive discussion of neural video compression methods (DCVC-RT, DCVC-DC, DCVC-FM, DVC, NVC) in Related Work, clarifying their autoencoder-based paradigm versus our INR-based approach.

---

> > ### Comment · Reviewer_dW3g · 2025-11-25
> >
> > W1: DCVC-RT also shares encoder and decoder weights across all videos. Only the latents (tokens) are video-specific. Both CAVINR and DCVC-RT train the encoder/decoder end-to-end. These portions seem identical to me. What am I missing?
> >
> > W2: I don't see how this work enables video archival or medical imaging compared to DCVC-RT given the significant gap in rate distortion.
> >
> > W3: Acknowledged.

---

> ### Author Response · Authors · 2025-11-24
>
> ## Response to Q1: Why (x,y,t) instead of t-Only?
>
> The shift to spatiotemporal coordinates is a core contribution:
>
> **Pixel-Level Control:** Convolutional architectures (FastNeRV) apply shared kernels across spatial locations, limiting pixel-specific control. Our coordinate-attentive approach establishes direct correspondences between video tokens and each spatial location, enabling precise reconstruction (demonstrated by 6-9 dB gains).
>
> **Downstream Task Support:** (x,y,t) queries enable arbitrary-resolution reconstruction, super-resolution, frame interpolation, and spatial cropping—capabilities unavailable with frame-level generation.
>
> ## Response to Q2: Why Not Compare with DCVC-RT?
>
> We have now added comprehensive comparisons with DCVC-RT:
>
> **Speed Comparison (Table 9):** CAVINR achieves 1.97× faster encoding (247 vs. 125 FPS on A100 for 1080p) but slower decoding (7.7 vs. 112.8 FPS)—reflecting different optimization targets.
>
> **Rate-Distortion Comparison (new Figure 7):** On HEVC class D sequences, CAVINR demonstrates competitive compression efficiency against traditional codecs:
>
> | Method | bpp @ ~33.5 dB | bpp @ ~35 dB | PSNR Range |
> | -- | -- | -- | -- |
> | HM-16.25 | 0.069 (33.52 dB) | 0.116 (35.32 dB) | 25.55–35.32 dB |
> | DCVC-FM | 0.036 (33.94 dB) | 0.061 (35.58 dB) | 26.14–38.60 dB |
> | DCVC-RT | 0.045 (33.83 dB) | 0.073 (35.05 dB) | 26.96–36.82 dB |
> | **CAVINR** | **0.062** (33.51 dB) | **0.105** (35.27 dB) | 30.78–36.29 dB |
>
> Key observations:
> - CAVINR outperforms HM-16.25 by 10.1% bitrate reduction at ~33.5 dB and 9.5% at ~35 dB
> - Specialized learned codecs (DCVC-FM, DCVC-RT) achieve higher compression efficiency, as expected given their end-to-end optimization for video compression
> - CAVINR offers unique INR capabilities (arbitrary resolution, partial decoding, frame interpolation) unavailable in autoencoder-based methods
>
> These results demonstrate that CAVINR maintains competitive rate-distortion performance against traditional codecs while offering unique INR capabilities that differentiate it from autoencoder-based learned codecs.
>
> ## Response to Q3: Applicability to Images and 3D Signals
>
> **Images:** CAVINR naturally extends to images by setting t=0. We added experiments at 512×512 and 1024×1024 resolutions (Table 5), where CAVINR-L achieves 30.7 dB and 30.2 dB PSNR, respectively, while ANR-L encounters OOM errors—demonstrating superior scalability.
>
> **3D Data:** Our method applies to 3D signals with (x,y,z) coordinates. The Axis-Adaptive Embedding may require adjustment for specific 3D data where dimensional frequency characteristics differ from video. We plan to extend this work to 3D domains in future research.

---

> > ### Comment · Reviewer_dW3g · 2025-11-25
> >
> > Q1. Do you demonstrate these downstream task capabilities in the paper? I am struggling to find them in the manuscript, I only see compression/reconstruction.
> >
> > Q2. What is meant by "end-to-end optimization for video compression?" What is CAVINR optimized for, if not compression? Reconstruction is trivial given a sufficiently large embedding, so is the task not always compression? In terms of the unique INR capabilities, where are these demonstrated? The paper seems to present only reconstruction/compression results at fixed resolutions.
> >
> > Q3. On the subject of Table 5, how can one perform 1024x1024 training/evaluation on K400, given most videos in that dataset are much smaller? In terms of 3D and single frame, acknowledged.

---

> > > ### Author Response · Authors · 2025-11-29
> > >
> > > ## Response to Q1 Follow-up: Downstream Task Demonstration
> > >
> > > We have added super-resolution experiments following the variational coordinate approach from ANR (Zhang et al., 2024, Section 3.2). By introducing controlled perturbations to input coordinates during inference, CAVINR performs zero-shot super-resolution without additional training.
> > >
> > > **New Table: Super-Resolution Results**
> > >
> > > | Methods | Resolution | #θ' | Epoch | GPU hrs | PSNR (Train/K400/SthV2/UCF101) | SSIM (Train/K400/SthV2/UCF101) |
> > > | -- | -- | -- | -- | -- | -- | -- |
> > > | ANR  | 128 | 27k | 150 | 96 | 33.1/33.2/33.2/33.6 | 0.933/0.919/0.934/0.929 |
> > > | CAVINR  | 128 | 27k | 50 | 8 | **33.3/33.3/33.4/33.8** | **0.935/0.926/0.936/0.935** |
> > > | ANR (SR 2×)  | 256 | 27k | 0 | 0 | 28.9/29.3/28.8/28.3 | 0.861/0.841/0.874/0.867 |
> > > | CAVINR (SR 2×)  | 256 | 27k | 0 | 0 | **31.1/30.1/31.5/31.8** | **0.919/0.907/0.916/0.913** |
> > >
> > > Key observations:
> > >
> > > - Models trained at 128×128, tested at 256×256 (2× super-resolution) with **zero additional training**
> > > - CAVINR achieves **+2.2 dB PSNR improvement** over ANR in super-resolution
> > > - This demonstrates the coordinate-based INR capability unavailable to autoencoder methods
> > >
> > > ## Response to Q2 Follow-up: Optimization Target Clarification
> > >
> > > The reviewer raises a valid point. To clarify:
> > >
> > > **CAVINR is optimized for reconstruction quality** (MSE loss), which corresponds to optimizing compression at a fixed bitrate determined by video token size.
> > >
> > > **Revised framing:**
> > >
> > > - DCVC-RT jointly optimizes rate-distortion with learned entropy models and variable-rate capability
> > > - CAVINR optimizes reconstruction quality at a fixed representation size (no entropy modeling)
> > >
> > > The phrase "end-to-end optimization for video compression" was imprecise. The key distinction is that CAVINR operates as a **fixed-rate representation** without entropy modeling, while DCVC-family methods are **variable-rate codecs** with sophisticated entropy models. This is a limitation of CAVINR compared to learned codecs.
> > >
> > > ## Response to Q3 Follow-up: High-Resolution Dataset Preparation
> > >
> > > We clarify our experimental protocol for 512×512 and 1024×1024 experiments:
> > >
> > > We apply standard data augmentation transforms to resize all videos to the target resolution:
> > >
> > > ```python
> > > transforms.Compose([
> > >     transforms.Resize(target_size),    # Resize shorter side
> > >     transforms.CenterCrop(target_size), # Center crop to square
> > >     transforms.ToTensor(),
> > > ])
> > > ```
> > >
> > > This approach:
> > >
> > > 1. Resizes the shorter side to the target resolution using bicubic interpolation
> > > 2. Applies center cropping to obtain square frames
> > > 3. Maintains consistent evaluation across all videos in the dataset
> > >
> > > We acknowledge that this involves upsampling for videos with native resolution below the target size. This is a common practice in video representation learning to ensure consistent input dimensions. We have add this preprocessing detail to the revised manuscript.

---

> ### Author Response · Authors · 2025-11-29
>
> ## Response to W1 Follow-up: Architectural Distinction
>
> Thank you for this important clarification question. We acknowledge the similarity you've identified and provide a more precise technical distinction:
>
> **Key Difference: Information Flow Path**
>
> | Aspect | DCVC-RT | CAVINR |
> | -- | -- | -- |
> | Reconstruction input | Quantized latent features → spatial upsampling | Explicit (x,y,t) coordinates + video tokens |
> | Spatial decoding | Feature-to-feature transformations | Coordinate-to-pixel mapping |
> | Query mechanism | Fixed spatial grid determined by latent resolution | Arbitrary coordinate queries |
>
> The fundamental distinction lies in **how pixel values are generated**:
>
> - **DCVC-RT**: Decodes through learned upsampling of compressed feature maps. The spatial structure is implicit in the feature arrangement.
> - **CAVINR**: Explicitly queries each (x,y,t) coordinate against video tokens via cross-attention to generate pixel values. This coordinate-based query mechanism is the defining characteristic of INR methods.
>
> We agree that both methods share weights across videos and train end-to-end. The distinction is not in weight sharing, but in the **reconstruction paradigm**: feature upsampling vs. coordinate-based function evaluation.
>
> ## Response to W2 Follow-up: Application Justification
>
> We provide a concrete example demonstrating CAVINR's advantage for medical imaging:
>
> **Ultrasound Video Compression**: In ultrasound imaging, the effective diagnostic content is contained within a **fan-shaped (sector) region**, while the surrounding rectangular frame contains non-diagnostic black regions. Traditional codecs like DCVC-RT must encode the entire rectangular frame.
>
> CAVINR's coordinate-based approach enables:
>
> - **Selective encoding**: Only encode coordinates within the fan-shaped ROI
> - **Selective decoding**: Reconstruct only the diagnostically relevant region
> - **Storage efficiency**: Avoid wasting bits on non-informative regions
>
> This capability is architecturally impossible for feature-upsampling methods, which must process the complete spatial grid.

---

### Comment · Area_Chair_x9wa · 2025-11-27

Dear Reviewers,

Author responses are now posted. Please add your discussion comment(s) and update score/confidence as needed. Thank you!

Best regards,

AC

---

### Note · Program_Chairs · 2026-01-17
**Submission Desk Rejected by Program Chairs**

The following references in this submission do not refer to real documents and/or have major errors in bibliographic information:

 Didier Le Gall and Ali Tabatabai. Learned video compression. In ICCV, pp. 3059-3068. IEEE, 2020.